# ER stress-induced mediator C/EBP homologous protein thwarts effector T cell activity in tumors through T-bet repression

Yu Cao [1], Jimena Trillo-Tinoco[1], Rosa A. Sierra[1], Carmen Anadon[1], Wenjie Dai[1], Eslam Mohamed[1], Ling Cen[2], Tara L. Costich[1], Anthony Magliocco[3], Douglas Marchion[3], Richard Klar[4], Sven Michel[4], Frank Jaschinski[4], Richard R. Reich[5], Shikhar Mehrotra[6], Juan R. Cubillos-Ruiz[7,8], David H. Munn[9], Jose R. Conejo-Garcia[1] & Paulo C. Rodriguez[1]

Understanding the intrinsic mediators that render CD8$^+$ T cells dysfunctional in the tumor microenvironment is a requirement to develop more effective cancer immunotherapies. Here, we report that C/EBP homologous protein (Chop), a downstream sensor of severe endoplasmic reticulum (ER) stress, is a major negative regulator of the effector function of tumor-reactive CD8$^+$ T cells. Chop expression is increased in tumor-infiltrating CD8$^+$ T cells, which correlates with poor clinical outcome in ovarian cancer patients. Deletion of Chop in T cells improves spontaneous antitumor CD8$^+$ T cell immunity and boosts the efficacy of T cell-based immunotherapy. Mechanistically, Chop in CD8$^+$ T cells is elevated primarily through the ER stress-associated kinase Perk and a subsequent induction of Atf4; and directly represses the expression of T-bet, a master regulator of effector T cell function. These findings demonstrate the primary role of Chop in tumor-induced CD8$^+$ T cell dysfunction and the therapeutic potential of blocking Chop or ER stress to unleash T cell-mediated antitumor immunity.

[1] Department of Immunology, H. Lee Moffitt Cancer Center & Research Institute, Tampa, FL 33612, USA. [2] Cancer Informatics Core, H. Lee Moffitt Cancer Center & Research Institute, Tampa, FL 33612, USA. [3] Department of Pathology, H. Lee Moffitt Cancer Center & Research Institute, Tampa, FL 33612, USA. [4] Secarna Pharmaceuticals GmbH & Co. KG, 82152 Planegg/Martinsried, Germany. [5] Biostatistics Program, H. Lee Moffitt Cancer Center & Research Institute, Tampa, FL 33612, USA. [6] Department of Surgery, Medical University of South Carolina, Charleston, SC 29425, USA. [7] Department of Obstetrics and Gynecology, Weill Cornell Medicine, New York, NY 10065, USA. [8] Sandra and Edward Meyer Cancer Center, Weill Cornell Medicine, New York, NY 10065, USA. [9] Department of Pediatrics, Georgia Cancer Center, Augusta University, Augusta, GA 30912, USA. Correspondence and requests for materials should be addressed to P.C.R. (email: Paulo.Rodriguez@Moffitt.org)

The impaired T cell function present in most patients and experimental animals with cancer is a primary mechanism for the evasion of protective antitumor immunity and represents a major limitation in the development of promising immunotherapies against cancer[1,2]. Different elements in the tumor microenvironment (TME), including immunosuppressive myeloid subsets; regulatory T cells; inflammatory cytokines; and components related with the metabolism of oxygen, glucose, or amino acids, play significant roles in the inhibition of antitumor T cell responses[3–5]. However, the molecular mediators and downstream mechanisms by which T cells become dysfunctional in tumors remain unclear. Under homeostatic conditions, self-inhibitory metabolic signals are evoked following T cell activation and expansion to control exaggerated T cell responses[6]. These include a deceleration of protein synthesis, inhibition of glycolytic potential, activation of apoptosis drivers, and expression of quiescence mediators[7–11]. Although these pathways are critical for the contraction of primed and expanded T cells, the role of these events in the evasion of protective T cell immunity in tumors remains incompletely understood.

The endoplasmic reticulum (ER) is an essential organelle that serves as the primary site for the production, assembly, and modification of proteins[12,13]. Hostile conditions in the TME, including hypoxia, high levels of reactive species, and nutrient deprivation, alter the ER homeostatic balance, leading to ER stress[14,15]. Elevated ER stress selects cancer cell clones with the ability to adapt to the harsh TME and correlates with tumor stage, aggressiveness, and low survival in patients with different malignancies[16,17]. In addition, increased ER stress has recently emerged as a major mediator of the immunosuppressive activity of tumor-associated myeloid cells[18,19]. To cope with ER stress, cancer-associated cells activate an integrated signaling network known as the unfolded protein responses (UPR), which is characterized by the cleavage of the activating transcription factor 6 (Atf6), priming of the inositol-requiring enzyme 1 alpha (Ire1α), and auto-phosphorylation of the protein kinase R (PKR)-like ER kinase (Perk, encoded by gene $Eif2ak3$)[20]. Cleaved Atf6 drives the expression of ER-associated degradation genes, whereas the active Ire1α induces mRNA splicing of X-box binding protein 1 (Xbp1) or/and Ire1α-dependent decay of ER-stress associated mRNAs. Moreover, self-activated Perk phosphorylates eIF2α, inducing integrated stress responses (ISR) that aim to decrease mRNA translation and protein loading into the ER, and to promote the expression of stress-related proteins such as the activating transcription factor 4 (Atf4)[21]. Additional ER stress-independent drivers of ISR include the heme-regulated eIF2α kinase (Hri, encoded by $Eif2ak1$), the protein kinase R (Pkr, encoded by $Eif2ak2$), and the general control non-derepressible 2 kinase (Gcn2, encoded by $Eif2ak4$)[22]. Although the induction of controlled UPR has been well established in malignant cells as a requirement for the adaptation to stress processes, it also regulates tumor cell death induced by several chemotherapy agents[23]. Also, despite the fact that the UPR signals regulate the activity of cancer cells and tumor-associated myeloid cells[24–26], the intrinsic effect of the UPR mediators in the dysfunction occurring in tumor-infiltrating T lymphocytes (TILs) remains unclear.

Upregulation of the transcription factor Chop, encoded by the $Ddit3$ gene, occurs in response to unbalanced ISR or exaggerated UPR and primarily initiates cellular apoptosis processes[27,28]. Notably, recent reports showed the effect of Chop in non-apoptosis-related cellular events[29]. In addition, previous findings indicated the role of Chop in the immunoregulatory function of tumor-associated myeloid-derived suppressor cells (MDSC)[19,30]. Deletion of Chop impaired MDSC immunosuppressive activity, thereby enhancing protective antitumor T cell responses.

Although Chop has emerged as a primary mediator of the tolerogenic activity of tumor-infiltrating myeloid cells, the direct role of Chop in antitumor CD8+ T cell immunity remains to be elucidated.

In this study, we sought to understand the endogenous effect of Chop in the impaired function of CD8+ T cells in solid malignancies. We demonstrate an intrinsic inhibitory role of Perk-induced Chop in tumor-infiltrating T cells. Accordingly, deletion or silencing of Chop potentiate cytotoxic T cell activity and overcome tumor-induced T cell dysfunction. These findings show for the first time the therapeutic potential of blocking Chop in CD8+ T cells, or its upstream driver Perk, as a strategy to restore protective T cell immunity against cancer and a platform to enhance the effectiveness of T cell-based immunotherapies.

## Results

**Chop in CD8+ TILs correlates with poor clinical responses**. We sought to determine whether CD8+ T cells upregulate Chop expression upon infiltration into the TME. Thus $Ddit3$ mRNA levels were assessed in CD8+ T cells sorted from the spleens of tumor-free mice or tumors and spleens of mice bearing subcutaneous (s.c.) 3LL, EL-4, MCA-38, or B16 cancer cells. Higher levels of $Ddit3$ mRNA were detected in sorted CD8+ TILs, compared to their splenic counterparts from tumor-bearing or tumor-free mice (Fig. 1a). In addition, a corresponding augmented expression of Chop, and a higher frequency of Chop+ cells, were noticed in CD8+ TILs from mice bearing B16 melanoma or 3LL lung carcinoma cells, as well as in ascites-related CD8+ T cells from ID8-$Defb29$/$Vegf-a$ ovarian tumors[31,32], compared to splenic CD8+ T cells from the same mice (Fig. 1b, Supplementary Fig. 1a, b). Interestingly, the upregulation of Chop in tumor-associated CD8+ T cells correlated with an enlargement of the ER as tested by ER tracker (Supplementary Fig. 1c). Next, we investigated whether a heightened expression of Chop was also occurring in human CD8+ TILs. Using a cohort of patients with advanced ovarian carcinoma and controls, we detected higher CHOP levels and augmented frequency of CHOP+ cells in CD8+ T cells infiltrating solid ovarian tumors, compared to autologous peripheral blood CD8+ T cells or lymphocytes from healthy controls (Fig. 1c, d, Supplementary Fig. 1d, e for gating strategies). Also, in agreement with our previous findings[19], we noticed an elevated expression of CHOP in tumor-infiltrating MDSC, which was more pronounced than that observed in CD8+ TILs (Supplementary Fig. 1e). Next, we validated the augmented expression of CHOP in human CD8+ TILs in 87 annotated cases of advanced high-grade serous epithelial ovarian cancer and 12 controls using high-resolution automated image acquisition analysis with 2 independent anti-CHOP antibodies. A broad elevated expression of CHOP was found in the ovarian tumors, compared to healthy tissues (Supplementary Fig. 2a, b). In addition, specific non-subjective analysis focusing on T cells showed higher levels of nuclear and cytosolic CHOP in tumor-related CD8+ T cells, compared to those from healthy ovary tissues (Fig. 1e, f, Supplementary Fig. 2a, c, d), suggesting the driving effect of the TME in the induction of CD8+ T cell CHOP. More importantly, the augmented expression of nuclear CHOP in CD8+ TILs was significantly associated with a decreased overall survival in patients with ovarian cancer (median survival 75 vs. 119 months for clone 9C8 and 64 vs. 104 for polyclonal Ab-R-20; Fig. 1g, Supplementary Fig. 2e). Moreover, the elevated levels of nuclear CHOP in CD8+ TILs correlated with suboptimal cytoreductive surgery or tumor debulking, a key event that predicts poor clinical responses in ovarian carcinoma patients (Fig. 1h)[33]. Interestingly, the increased levels of CHOP in the cytosol of CD8+

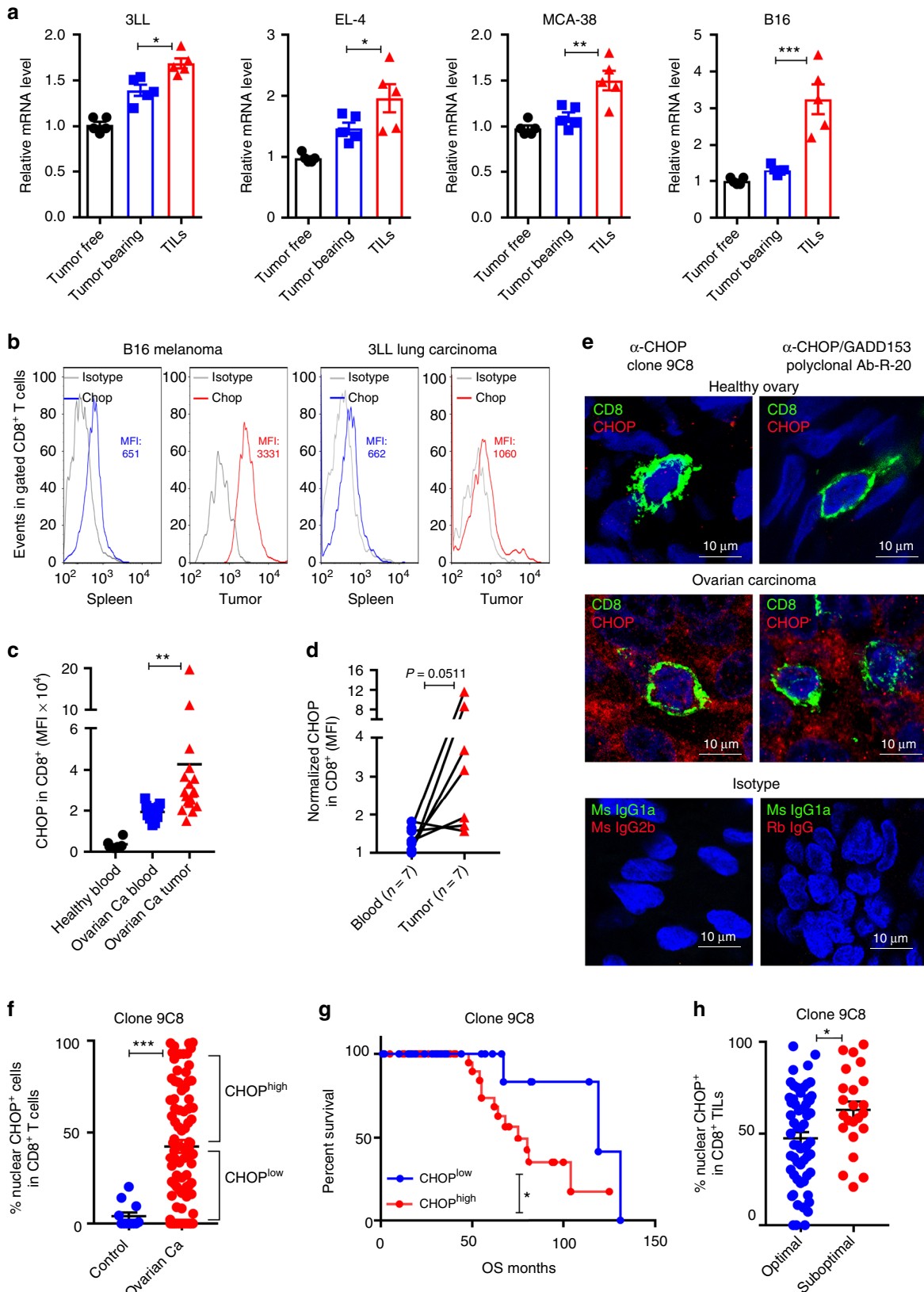

TILs were associated with unsuccessful tumor debulking but did not significantly correlate with changes in overall survival (Supplementary Fig. 2f, g). Thus, the increased expression of nuclear CHOP in CD8$^+$ TILs is a key predictor of poor clinical responses in advanced ovarian cancer.

**Primed Perk controls the expression of Chop in CD8$^+$ T cells.** The process of T cell expansion upon T cell receptor engagement is characterized by a significant increase in protein synthesis and secretory demands, which trigger ER stress[34–36]. Since most of the TILs show transcript patterns associated with activation[37], we

**Fig. 1** Increased Chop in CD8+ tumor-infiltrating T lymphocytes (TILs) correlates with poor survival in ovarian cancer. **a** *Ddit3* mRNA levels in tumor-associated CD45+ CD8+ T cells (TILs) sorted from subcutaneous 3LL, EL-4, MCA-38, or B16 tumors and CD8+ T cells from the spleens of the same tumor-bearing mice (Tumor bearing) or tumor-free mice (Tumor free). Bar graphs show the mean ± s.e.m. ($n = 5$ mice/group). **b** Chop expression in CD8+ TILs from B16 melanoma tumors (left) and 3LL tumors (right), compared with splenic CD8+ T cells from the corresponding tumor-bearing mice. Chop was detected by fluorescence-activated cell sorter and levels indicated by mean fluorescence intensity (MFI). Representative findings from four repeats. **c** CHOP in CD8+ TILs from ovarian carcinoma patients (Ovarian Ca tumor, $n = 18$) compared to peripheral blood CD8+ T cells from ovarian carcinoma patients (Ovarian Ca blood, $n = 11$) or healthy controls (Healthy blood, $n = 6$). **d** CHOP levels in autologous CD8+ TILs (Tumor) and peripheral blood CD8+ T cells (Blood) from ovarian carcinoma patients ($n = 7$). **e** Representative image (scale 10 μm) showing CHOP expression in CD8+ TILs from ovarian carcinoma patients compared to CD8+ T cells from healthy ovarian tissues. Isotype (red) or CHOP (red, tested by clone 9C8 (left) or polyclonal antibody R-20 (right)), CD8 (green) and DAPI (blue) were detected by confocal microscopy. **f** Percentage of nuclear CHOP+ cells (clone 9C8) among tumor-associated CD8+ T lymphocytes in a tissue microarray containing advanced ovarian carcinoma tissues ($n = 87$) vs. normal ovarian tissues ($n = 12$). **g** Overall survival in advanced ovarian tumor patients having increased frequency of nuclear CHOP in CD8+ TILs (CHOP[high]) ($n = 52$) vs. those having low frequency of nuclear CHOP in CD8+ TILs (CHOP[low]) ($n = 29$) (logrank 4.39, $p = 0.0361$ using Gehan–Breslow–Wilcoxon test; cutoff was established as described in the Methods section). Studies were developed using the anti-CHOP antibody clone 9C8. **h** Percentage of CD8+ TILs having nuclear CHOP (clone 9C8) in ovarian cancer patients that had optimal ($n = 59$) vs. suboptimal ($n = 23$) cytoreductive debulking surgery. Bar graphs represent mean value ± s.e.m., $*p < 0.05$, $**p < 0.01$, $***p < 0.001$ were calculated using two-tailed unpaired Student's $t$ test

determined whether Chop is induced after T cell stimulation. A time-dependent induction of Chop was observed in anti-CD3/CD28-stimulated mouse and human T cells (Fig. 2a, Supplementary Fig. 3a, b) and in antigen-specific CD8+ T cells from OT-1 or Pmel mice activated with the corresponding peptide (Supplementary Fig. 3c). Moreover, elevated levels of Chop and higher frequency of Chop+ cells were detected in Pmel CD8+ T cells previously transferred into mice that received vaccination with gp100[25–33] peptide, compared to those from non-vaccinated controls (Fig. 2b). In addition, we noted higher Chop levels in proliferating transferred Pmel T cells from gp100[25–33]-vaccinated mice (activation-driven T cell proliferation) compared to non-vaccinated cohorts (homeostatic T cell division) (Supplementary Fig. 3d), suggesting the increased expression of Chop under activation-induced CD8+ T cell proliferation.

Next, we aimed at elucidating the role of the ER stress and UPR signaling as mediators of the Chop upregulation in primed T cells. ER stress inhibitor Tauroursodeoxycholic acid (Tudca)[38] impaired the induction of Chop in activated CD8+ T cells, whereas treatment with the ER stress inducer Thapsigargin[39] enhanced it (Fig. 2c). Accordingly, we found a significant increase in the expression and activation of the major UPR mediators, Perk and Ire1α, as well as an upregulation of the *Ddit3* mRNA-inducing factor Atf4 in stimulated mouse and human CD8+ T cells, compared to non-activated controls (Fig. 2d, Supplementary Fig. 4a). Also, a similar UPR activation was detected after priming of antigen-specific Pmel or OT-1 CD8+ T cells with the corresponding peptides gp100[25–33] or OVA[257–264], respectively (Supplementary Fig. 4b). Because the induction of Chop in cancer cells undergoing ER stress has been related with the activation of Perk and a subsequent upregulation of Atf4[40], we analyzed the effect of Perk in the expression of Chop and Atf4 in stimulated T cells. For this, we used CD8+ T cells from T cell-conditional *Eif2ak3*-null mice (*Eif2ak3*[T cell-KO]), developed after crossing *Eif2ak3*[flox/flox] mice and mice harboring *CD4*-driven Cre recombinase. A significant decrease in Chop and *Atf4* was noticed in activated *Eif2ak3*-deficient CD8+ T cells compared to controls (Fig. 2e, Supplementary Fig. 4c). In addition, silencing of *Atf4* in stimulated CD8+ T cells, using a pool of specific small interfering RNAs (siRNA; Supplementary Fig. 4d), impaired the accumulation of *Ddit3* mRNA (Supplementary Fig. 4e), indicating the mechanistic role of the Perk-dependent Atf4 induction in the upregulation of Chop in T cells. Next, we investigated whether the kinases that induce ISR independently of ER stress participated in the induction of Chop in activated T cells. Similar to the upregulation of Perk, we noticed a time-dependent induction of Hri, Pkr, and Gcn2 in stimulated CD8+ T cells,

compared to non-activated controls (Supplementary Fig. 4f). Next, we individually silenced the ISR-inducing kinases in activated CD8+ T cells using specific siRNA pools (Supplementary Fig. 4g). Knockdown of *Eif2ak3* had the highest effect in the prevention of *Ddit3* mRNA expression in primed CD8+ T cells, whereas silencing of *Eif2ak2* or *Eif2ak4* only showed a moderate impact (Supplementary Fig. 4h), confirming the primary role of the activation of ISR by the ER stress-related kinase Perk in the expression of Chop in stimulated T cells.

Next, we determined the role of Perk in the expression of Chop in CD8+ TILs. Lower levels of Chop were found in B16-associated CD8+ TILs from *Eif2ak3*[T cell-KO] mice, compared to CD8+ TILs from controls (Fig. 2f). To further understand the tumor-related pathways leading to the activation of Perk and the induction of Chop in CD8+ T cells, we focused on the effect of tumor-induced reactive oxygen species (ROS)[19]. An augmented ROS accumulation, phosphorylation of Perk, and induction of Chop were observed in primed T cells cultured in the presence of cell-free ovarian cancer ascites (Fig. 2g, h, Supplementary Fig. 5a–c). Also, inhibition of the ROS-related effects upon treatment with the ROS scavenger, L-NAC, prevented the phosphorylation of Perk and the induction Chop in ascites-treated primed CD8+ T cells (Fig. 2i, Supplementary Fig. 5d). These results indicate the primary role of ROS-induced Perk activation in the upregulation of Chop in tumor-exposed CD8+ T cells.

**Chop negatively regulates effector T cell responses**. To elucidate the effect of the deletion of Chop in the function of CD8+ T cells, we performed whole transcriptome analysis (RNA-Seq) in gp100[25–33]-primed wild-type and *Ddit3*-deficient Pmel T cells. We detected significant changes in 570 transcripts (232 increased vs. 338 decreased) in *Ddit3* null Pmel T cells compared to controls (Supplementary Fig. 6a, b). In addition, gene set enrichment analysis (GSEA) indicated that the deletion of *Ddit3* in primed CD8+ T cells promoted the expression of transcripts related with effector activity, activation, proliferation, interleukin-2 signaling, survival, glycolysis, and oxidative phosphorylation metabolism (Fig. 3a, b, Supplementary Fig. 6c). Accordingly, we found faster proliferation, augmented ability to respond to antigen stimulation, elevated levels of the cytotoxic mediator granzyme B, and higher interferon-γ (IFNγ) production in activated CD8+ T cells from *Ddit3*-deficient mice, compared to controls (Fig. 3c–e). Also, a slight decrease in the spontaneous and staurosporine-enforced expression of the apoptosis marker Annexin V was detected in activated *Ddit3* null CD8+ T cells, compared to controls

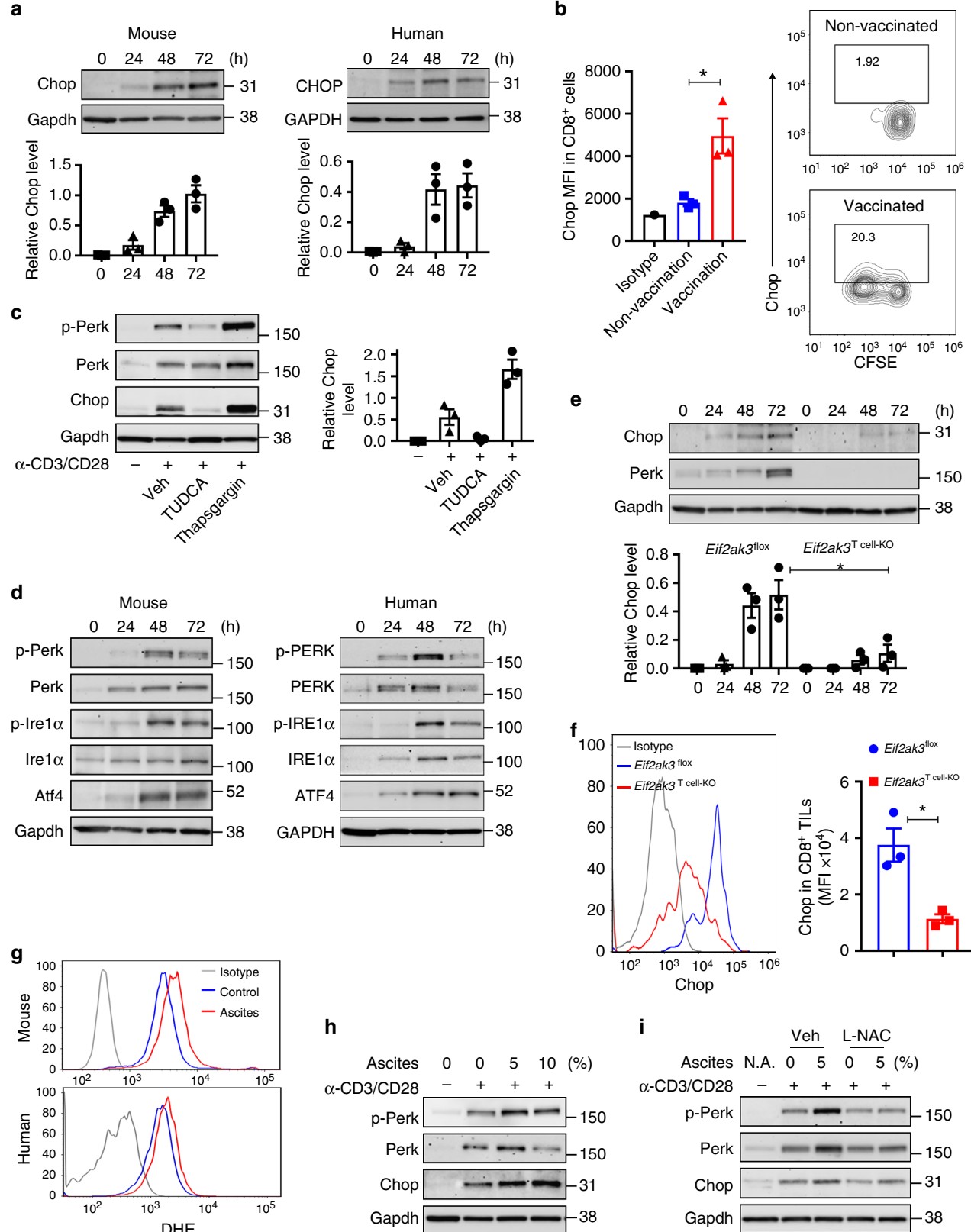

(Supplementary Fig. 6d, e). Moreover, in comparison to wild-type CD8[+] T cells, we observed that stimulated *Ddit3*-deficient T cells had an augmented extracellular acidification rate (ECAR) and oxygen consumption rate (OCR), suggesting a superior metabolic fitness through both an elevated glycolytic potential and mitochondrial oxidative phosphorylation, respectively

(Fig. 3f, g). Notably, in agreement with the upstream role of the Perk activation in response to ER stress in the Chop-mediated inhibitory effects, we found augmented IFNγ production in stimulated T cells treated with the ER stress inhibitor Tudca or lacking *Eif2ak3* (Fig. 3h, i). Next, we focused on the effects induced by the *Ddit3* deletion in the cytotoxic capacity of CD8[+]

**Fig. 2** Perk regulates Chop expression in primed CD8[+] T cells and CD8[+] TILs. **a** Upper panel: Time-dependent induction of Chop in murine (left) and human (right) T cells primed in vitro. T cells were stimulated with anti-CD3/CD28 and collected at the indicated time points (0–72 h). Lower panel: Densitometry quantitation of immunoblots (n = 3). **b** Carboxyfluorescein succinimidyl ester-labeled Pmel CD8[+] T cells (CD90.1[+]) were transferred into wild-type mice (CD90.2[+]), followed or not by vaccination with gp100$_{25-33}$ plus IFA. On day 4, Chop levels by mean fluorescence intensity (MFI) (left) and percentage of Chop[+] cells (right) were established by fluorescence-activated cell sorter (FACS) in gated CD8[+] CD90.1[+] T cells from lymph nodes of vaccinated vs. non-vaccinated mice (n = 3). **c** Left: Tauroursodeoxycholic acid (0.5 mM, at time 0) or Thapsigargin (100 nM, after 48 h) were added to CD8[+] T cells stimulated with plate-bond anti-CD3/CD28. Protein extracts were collected 72 h post-activation. Right: Quantitation of immunoblots (n = 3). **d** Unfolded protein response mediators Perk, Ire-1α, and Atf4 in stimulated murine (left) and human (right) T cells (0–72 h). Representative Immunoblot from n = 3. **e** CD8[+] T cells from *Eif2ak3*[flox] or *Eif2ak3*[T cell-KO] mice were primed with anti-CD3/CD28 for 0–72 h. Upper panel: Chop and Perk detected by immunoblotting; lower panel: Densitometry quantitation of immunoblots (n = 3). **f** Chop MFI levels in CD8[+] TILs from B16-bearing *Eif2ak3*[flox] and *Eif2ak3*[T cell-KO] mice (n = 3). Chop was detected in viable CD45[+]CD8[+] TILs by FACS (left) and MFI values compiled (right). **g** Representative reactive oxygen species levels in murine and human DHE-labeled CD8[+] T cells activated with anti-CD3/CD28 and treated for 24 h with 5% cell-free ovarian ascites obtained from mice bearing ID8-*Defb29*/*Vegf-a* ovarian tumors or 5% primary ascites from patients with ovarian cancer, respectively (n = 4). **h** Murine CD8[+] T cells were primed with anti-CD3/CD28 for 48 h and then cultured with or without ID8-*Defb29*/*Vegf-a* cell-free ovarian tumors ascites for 24 h (n = 3). **i** CD8[+] T cells treated as in **h** were cultured in the presence or the absence of 2 mM L-NAC during the ascites treatment (n = 3). Bar graphs represent mean ± s.e.m., *p < 0.05 was calculated using two-tailed unpaired Student's t test

T cells. A higher cytotoxic activity against gp100$_{25-33}$-loaded EL-4 cells was displayed by primed *Ddit3* null Pmel CD8[+] T cells, compared to that triggered by controls (Fig. 3j). Also, an elevated polarization into a CD44[high] CD62L[−] effector memory phenotype was noted in activated *Ddit3* knockout CD8[+] T cells, compared to controls (Fig. 3k, Supplementary Fig. 6f). Together, these results illustrate the negative role of Chop in multiple T cell-associated functions, including effector responses.

**Chop limits IFNγ production by inhibiting *Tbx21* transcription.** Seminal studies showed the key role of the transcription factor T-bet in the effector function of CD8[+] T cells[41,42]. Therefore, we tested a potential mechanistic interaction between Chop and T-bet in CD8[+] T cells. Increased levels of *Tbx21* (T-bet encoding gene) mRNA and T-bet protein were found in activated *Ddit3* null CD8[+] T cells, compared to wild-type counterparts (Fig. 4a). Conversely, *Ddit3* deletion in CD8[+] T cells did not impact the protein expression of other key transcription factors regulating T cell effector function, including the inhibitor of DNA binding 2 (Id2)[43], the positive regulatory domain I-binding factor (Blimp1)[44], and Eomesodermin (Eomes)[45] (Supplementary Fig. 7a). Also, consistent with the negative effect of Chop in the expression of T-bet, retroviral-driven expression of *Ddit3* in stimulated T cells resulted in lower levels of T-bet and IFNγ (Fig. 4b, Supplementary Fig. 7b). Furthermore, the elevated expression of T-bet in *Ddit3* null CD8[+] T cells correlated with a higher binding of T-bet to a consensus DNA sequence and an augmented expression of the T-bet target genes, *Ifng*, *Il12b2*, *Cbfa3*, and *Cxcr3* (Fig. 4c, Supplementary Fig. 7c), indicating the effect of Chop in T-bet signaling. Next, we studied whether Chop binds to the promoter region of *Tbx21*, impairing its transcriptional activity. Using the ChampionChIP database, we located a potential consensus Chop-binding sequence in the *Tbx21* promoter (chromosome 11: 97098007–97115331) (Fig. 4d). Accordingly, chromatin immunoprecipitation (ChIP) assays showed the endogenous binding of Chop to the predicted *Tbx21* promoter sequence in primed CD8[+] T cells (Fig. 4e). To validate the regulation of the *Tbx21* promoter by Chop, we established a dual luciferase system combining: (1) a plasmid harboring SV-40 promoter fused to 2× repeats of the Chop response element on the *Tbx21* gene (2x-CRE) driving Firefly luciferase (Supplementary Fig. 7d); and (2) a plasmid containing SV-40 promoter-driven Renilla luciferase. Both plasmids were co-transfected into 293T cells in combination with constructs expressing control or *Ddit3* sequences. A significant decrease in 2x-CRE T-bet luciferase activity was found after overexpression of *Ddit3* but not after co-transfection with the control vector (Fig. 4f), indicating

the inhibitory effect of Chop on *Tbx21* promoter activity. To confirm that Chop represses *Txb21* transcription through occupancy of its promoter, we established a competition chimeric decoy model in which a retrovirus containing 8x-CRE repeats on the *Tbx21* promoter (Supplementary Fig. 7e) was transduced into primed Chop-expressing CD8[+] T cells. Transduction with the 8x-CRE decoy insert increased the expression of T-bet in primed T cells expressing Chop (Fig. 4g), indicating the direct binding of Chop to the T-bet promoter. To further understand the role of a decreased T-bet in the effects induced by Chop upregulation, we bypassed the potential deleterious effects that the T-bet knock-down could induce in T cells by ectopically expressing T-bet in Chop-overexpressing CD8[+] T cells (Supplementary Fig. 7f). *Tbx21* transduction restored the production of IFNγ in Chop-overexpressing CD8[+] T cells (Fig. 4h), demonstrating the mechanistic and functional interaction between Chop and T-bet in CD8[+] T lymphocytes.

**Deletion of *Ddit3* in T cells promotes antitumor immunity.** We aimed to test whether the deletion of Chop in T cells promotes antitumor immunity. Thus, T cell conditional Chop null mice (*Ddit3*[T cell-KO]) were generated by crossing *Ddit3*[flox/flox] and CD4-Cre mice. Conditional deletion of *Ddit3* in T cells did not alter the distribution of CD4[+] and CD8[+] T cells in the spleen or peripheral blood (Supplementary Fig. 8a), indicating no obvious defects in T cell homeostasis. Also, in agreement with the inhibitory role of Chop in tumor-reactive T cells, a significant delay in B16 tumor growth was found in *Ddit3*[T cell-KO] mice compared to controls (Fig. 5a). Additionally, a similar augmented antitumor effect and an extension of survival were observed in *Ddit3*[T cell-KO] mice bearing MCA-38 or ID8-*Defb29*/*Vegf-a* tumors, respectively (Supplementary Fig. 8b, c). Furthermore, although equivalent percentages of CD8[+] T cells were found in tumors from *Ddit3*[T cell-KO] and control mice (Supplementary Fig. 8d), increased frequency of CD8[+] TILs expressing the effector mediators IFNγ and tumor necrosis factor-α (TNFα), the antigen-primed and activation markers CD44[+] CD69[+], the late effector T cell markers KLRG1[high] CD127[low], the effector differentiation driver T-bet, and the tissue resident memory markers CD69[+] CD103[+] were found in tumors from *Ddit3*[T cell-KO] mice, compared with *Ddit3*[flox/flox] controls (Fig. 5b–e, Supplementary Fig. 8e, f). To confirm the direct role of CD8[+] T cells in the antitumor effects noted in *Ddit3*[T cell-KO] mice, we depleted CD8[+] T cells using specific antibodies. Elimination of CD8[+] T cells restored tumor growth in B16-bearing *Ddit3*[T cell-KO] mice (Fig. 5f), confirming the role of CD8[+] T cells in the antitumor responses induced by

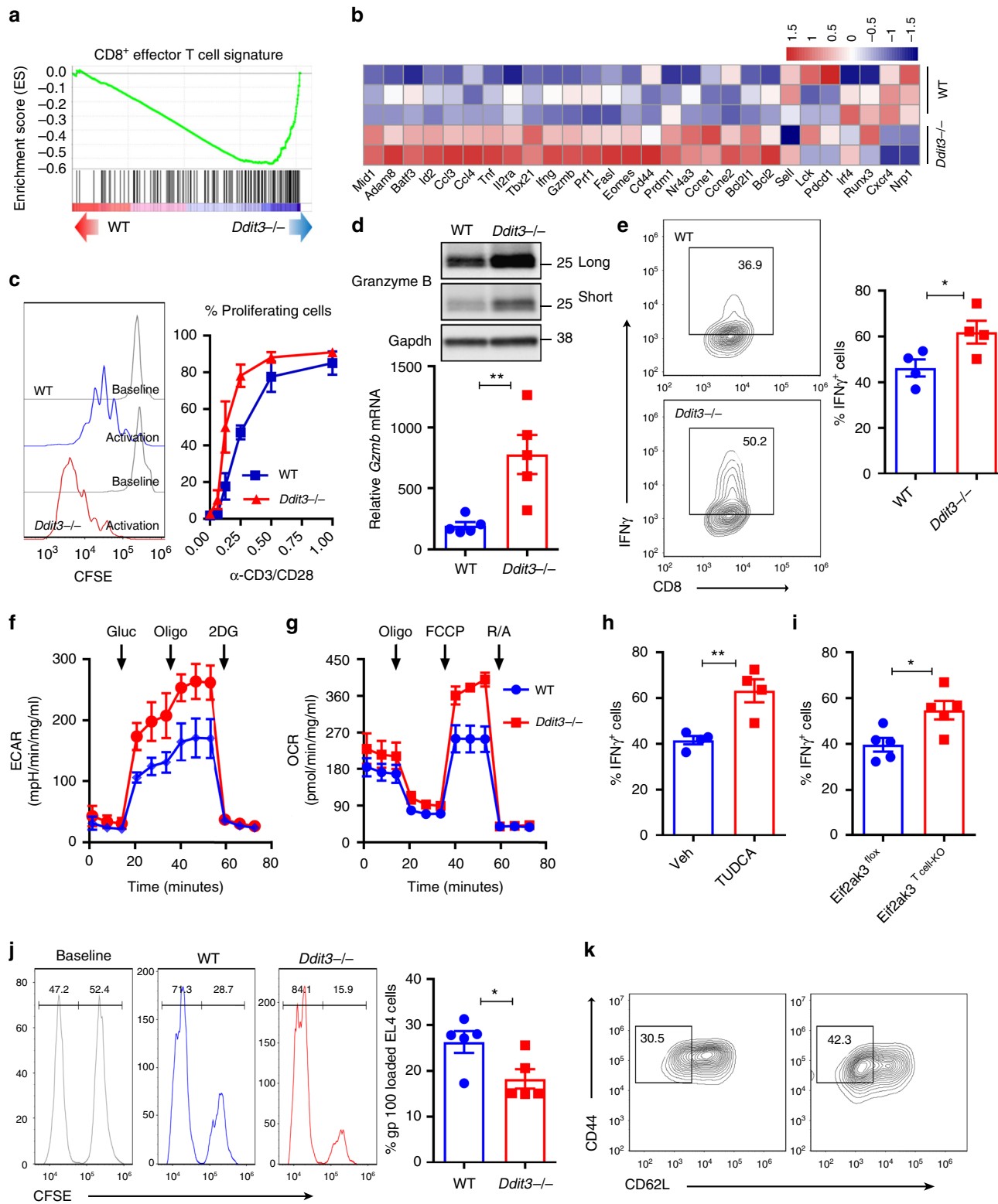

*Ddit3* deletion in T cells. Also, in accordance with the upstream role of Perk in the induction of Chop, a delayed tumor growth was found in *Eif2ak3*[T cell-KO] mice (Fig. 5g). Thus, our results show that deletion of Chop or Perk in T cells triggers spontaneous antitumor CD8[+] T cell activity.

**_Ddit3_ deletion increases the effect of T cell immunotherapy**. To address the effect of the knockdown of *Ddit3* in tumor-induced

CD8[+] T cell tolerance, we used an adoptive cellular therapy model against the melanoma tumor antigen gp100, in which pre-activated anti-gp100$_{25-33}$ transgenic CD90.1[+] Pmel T cells were transferred into CD90.2[+] congenic mice bearing established B16 melanoma tumors (Fig. 6a). Adoptive transfer of *Ddit3* null CD8[+] T cells induced a higher antitumor effect, compared to that triggered by Pmel controls (Fig. 6b), which correlated with a higher frequency of the transferred Pmel T cells expressing IFNγ

**Fig. 3** Chop negatively regulates effector CD8$^+$ T cell activity. **a** Gene set enrichment analysis was performed to determine the specific enrichment in effector CD8$^+$ T cell gene signature in primed wild-type or *Ddit3*−/− Pmel CD8$^+$ T cells. **b** Heatmap showing the expression of selective effector function-related transcripts in primed wild-type vs. *Ddit3*−/− Pmel CD8$^+$ T cells, as described in the Methods section. **c** Carboxyfluorescein succinimidyl ester (CFSE)-labeled wild-type or *Ddit3*−/− CD8$^+$ T cells were stimulated with anti-CD3/CD28 and proliferation tested after 72 h by fluorescence-activated cell sorter (FACS). Left: representative histogram of T cell proliferation (n = 6); right: T cell proliferation rates under different concentrations of anti-CD3/CD28 (µg/ml) (n = 6). **d** Granzyme B protein (upper panel, long- and short-time exposure) and *Gzmb* mRNA levels (lower panel) in wild-type or *Ddit3*−/− CD8$^+$ T cells primed with anti-CD3/CD28 for 72 h (n = 5). **e** Percentage of IFNγ$^+$ cells in wild-type or *Ddit3*−/− CD8$^+$ T cells primed as in (**d**). Right: representative FACS findings; left: merged percentage values from n = 4. **f** Extracellular acidification rate (ECAR) of wild-type or *Ddit3*−/− CD8$^+$ T cells primed as in **d** upon glycolysis stress analysis (n = 3). **g** Oxygen consumption rate (OCR) of wild-type or *Ddit3*−/− CD8$^+$ T cells activated as in **d** after mitochondrial stress analysis (n = 3). **h**, **i** Percentage of IFNγ$^+$ cells in CD8$^+$ T cells primed in the presence of Tauroursodeoxycholic acid (n = 4) (**h**) or from *Eif2ak3*$^{flox}$ or *Eif2ak3*$^{T cell-KO}$ mice (n = 5) (**i**). **j** In vitro cytotoxicity of wild-type and *Ddit3*−/− CD8$^+$ T cells was assessed by measuring EL-4 cell proportion after co-culturing gp100$_{25−33}$ pre-activated wild-type or *Ddit3*−/− Pmel CD8$^+$ T cells with EL-4 cells loaded with gp100$_{25−33}$ or control peptide (high or low CFSE, respectively). Cytotoxicity was evaluated after 24 h of co-culture by FACS. Right: representative FACS result; left: merged results from n = 5. **k** Representative result of CD44$^{high}$ CD62L$^-$ effector memory cells in wild-type or *Ddit3*-null CD8$^+$ T cells primed as in **d** (n = 10). Bar graphs represent mean ± s.e.m., *p < 0.05, **p < 0.01 were calculated using two-tailed unpaired Student's t test

in tumors and elevated production of IFNγ after challenging of splenocytes with gp100$_{25−33}$ peptide (Fig. 6c, d). Also, consistent with the regulatory role of Chop on effector T cells, we found an increased frequency of T-bet$^+$ cells in the transferred *Ddit3* null CD8$^+$ T cells infiltrating tumors, compared to TILs from transferred controls (Fig. 6e). Thus, results show the key role of Chop in tumor-induced T cell tolerance and suggest the benefit of targeting Chop in CD8$^+$ T cells as a platform to boost the efficacy of T cell immunotherapy.

To validate the therapeutic potential of inhibiting Chop in antitumor T cells, we next determined the effect of silencing Chop expression in CD8$^+$ T cells using specific locked nucleic acid (LNA)-Gapmer anti-sense oligonucleotide (ASO). A significant decrease in Chop expression was noted after unassisted treatment of primed T cells with ASO-*Ddit3*, compared to ASO-Control (Supplementary Fig. 9a). In addition, silencing of *Ddit3* increased the production of *Tbx21*, *Gzmb*, and *Ifng* to a similar extent of that noted in *Ddit3* null T cells (Supplementary Fig. 9b). Next, we tested the effect of pre-silencing of *Ddit3* in Pmel T cells transferred into B16 tumor-bearing mice. Briefly, Pmel T cells were activated with gp100$_{25−33}$ peptide in the presence of ASO-*Ddit3* or ASO-Control for 48 h, after which they were exposed to a second round of ASO and transferred into B16-bearing mice. Higher antitumor effects were observed after adoptive transfer of ASO-*Ddit3*-exposed Pmel T cells, compared to those induced by ASO-Control-conditioned Pmel T cells (Fig. 6f). Also, a higher frequency of CD8$^+$ T cells expressing IFNγ was found in tumor and spleen of mice receiving Pmel T cells treated with ASO-*Ddit3* (Fig. 6g, h). These results suggest that treatment of T cells with ASO-*Ddit3* increases the efficacy of T cell-based therapies by overcoming T cell dysfunction.

## Discussion
Our results reveal a new role of Chop as an intrinsic mediator of T cell dysfunction in tumors and suggest the therapeutic potential of inhibiting Chop or the UPR mediator Perk in CD8$^+$ T cells as a platform to overcome tumor-induced T cell suppression and a strategy to boost the efficacy of T cell-based immunotherapies.

CD8$^+$ T cell immunity is inhibited in the TME through different pathways including the elevated production of reactive oxygen and nitrogen species, the depletion of nutrients including glucose, arginine, and tryptophan, the limitation of oxygen availability, and the acidification of the extracellular milieu. However, the molecular signals by which this stressful and hostile milieu prevents effective T cell responses remain practically unknown. Our results show the induction of Chop in CD8$^+$ TILs as a major mechanism of inhibition of antitumor immunity. In

addition to the expression of Chop in CD8$^+$ TILs, we found a paradoxical induction of UPR and Chop in proliferating T cells, suggesting that tumors hijack a homeostatic pathway that negatively regulates T cell reactivity as a mechanism of immune evasion. In normal activated T cells, the induction of ER stress is the result of the increased secretory capacity occurring as part of T cell expansion and differentiation[34,35]. In agreement, modulation of UPR signaling through deletion of the ER chaperone gp96 or Xbp1 impaired T cell differentiation into effector populations[36,46]. Notably, the decreased secretory capacity present in CD8$^+$ T cells in tumors[47] rules out its role as a driver of UPR in TILs. Instead, our results showed the role of the ROS-dependent activation of Perk in the induction of Chop in tumor-exposed T cells. Upregulation of Chop was also found in primed T cells exposed to IDO-expressing dendritic cells or cultured under tryptophan deprivation conditions, which was mediated through the ISR-related kinase GCN2[48,49]. Additional studies showed that mimicking the effects triggered by amino acid starvation or deletion of amino acid transporters resulted in the activation of Chop in T cells[50,51]. Interestingly, similar signals were found to trigger ER stress in tumor-associated myeloid cells, driving their immunosuppressive activity[18,19,24–26]. Although our results show higher overall levels of Chop in tumor MDSC, the upregulation of Chop in CD8$^+$ TILs intrinsically regulated their antitumor activity. Thus, tumor-associated cellular stress pathways activate UPR signaling in myeloid cells and T cells as a master signal of evasion of protective antitumor immunity. Since ER stress additionally regulates survival of cancer cells, it is possible that the inhibition of ER stress or major UPR mediators could induce significant antitumor and immunogenic effects in individuals with tumors[52]. Development of these clinical studies remains to be tested and could have a major impact in the improvement of therapies in cancer.

Adoptive T cell transfer immunotherapies represent a promising treatment for patients with solid tumors[53]. However, only a fraction of the patients undergoing these therapies reach long-term clinical responses[54]. A possible explanation for the uneven efficacy of these treatments is the inability of the transferred T cells to accomplish tumor killing. We report an approach to boost the activity of transferred antitumor CD8$^+$ T cells by inhibiting ER stress or silencing Chop during the expansion phase. This could be of particular benefit for tumor therapies based on the expansion of tumor-infiltrating T cells or T cells transduced with chimeric antigen receptors. Chop deletion increased various signals associated with CD8$^+$ T cell reactivity, including higher expression of cytotoxic, proliferation, survival, metabolism, and activation mediators. Mechanistically, we show that Chop directly blunts *Tbx21* transcription, thereby impairing

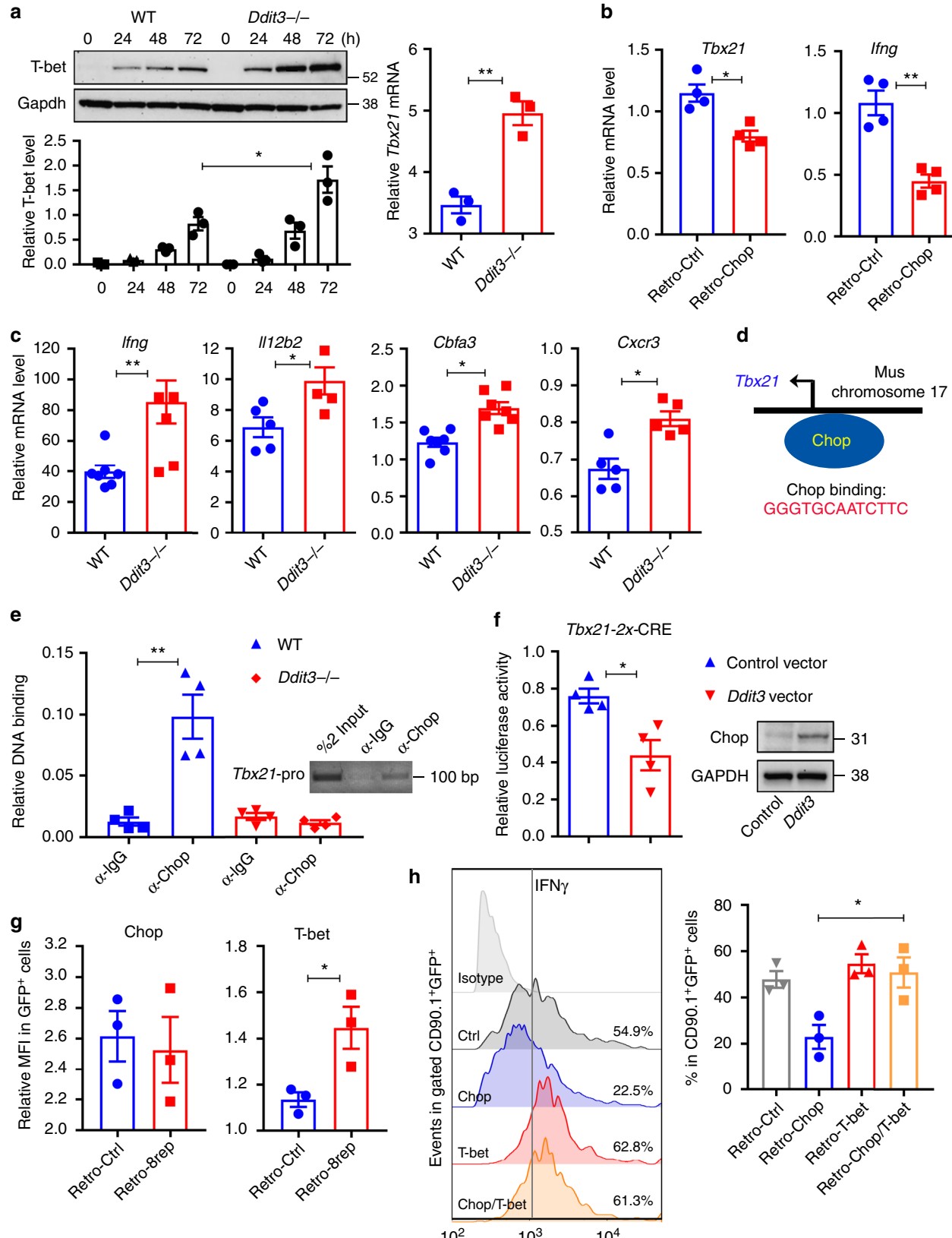

T cell effector immunity. Because Chop expression has been typically linked to extreme stress responses and activation of pro-apoptotic transcripts, we cannot rule out a transcriptional anti-apoptotic effect after Chop deletion in T cells. However, the assertion that Chop exclusively functions as a nuclear transcription factor is an oversimplification of the dynamic regulation of Chop during cellular stress[55,56]. In fact, complex roles and regulation of Chop, including both cytoplasmic and nuclear distribution, have been previously described[55–58]. The mechanisms of T cell regulation by cytosolic and nuclear Chop and the

**Fig. 4** Chop negatively regulates T-bet expression. **a** Time-dependent expression (upper panel) and corresponding densitometry quantitation (lower panel) of T-bet in primed wild-type and *Ddit3−/−* CD8+ T cells. Left: protein level (0–72 h); right: *Tbx21* mRNA levels 48 h post-activation. CD8+ T cells were stimulated with plate-bound anti-CD3/CD28 (n = 3). **b** *Tbx21* and *Ifng* mRNA expression in activated CD8+ T cells infected with control retrovirus (Retro-Ctrl) or *Ddit3*-expressing retrovirus (Retro-Chop). Cells were primed for 48 h and then infected for additional 48 h in the presence of the stimulating anti-CD3/CD28 antibodies (n = 4). **c** *Ifng*, *Il12b2*, *Cbfa3*, and *Cxcr3* mRNA levels in control vs. *Ddit3−/−* CD8+ T cells primed as in **a** (n = 5). **d** Predicted Chop-binding site in the *Tbx21* promoter region (GGGTGCAATCTTC). **e** Chromatin immunoprecipitation assay for the endogenous binding of Chop to *Tbx21* promoter in primed wild-type or *Ddit3−/−* CD8+ T cells. Chop-binding activity was measured by real-time quantitative PCR, compared with IgG binding activity after normalizing to the activity of anti-H3 (n = 4). **f** A dual luciferase system composed of 2x-CRE containing Firefly luciferase reporter and the control Renilla luciferase reporter was transfected into 293T cells in combination with *Ddit3*-expressing or control vectors. n = 4 experimental repeats. **g** Expression of Chop (left) and T-bet (right) by fluorescence-activated cell sorter (FACS) upon transduction of primed CD8+ T cells with green fluorescent protein (GFP)-coding retroviruses containing control or 8x-CRE sequences. Cells were primed for 48 h and then infected for another 48 h in the presence of the stimulating anti-CD3/CD28 antibodies plus interleukin (IL)-2 (50 U/ml). n = 3 independent repeats. **h** Interferon-γ (IFNγ) levels in primed CD8+ T cells transduced with: (1) GFP/CD90.1-expressing control virus (Ctrl); (2) Chop/CD90.1-expressing virus and GFP-expressing control virus (Chop); (3) CD90.1-expressing control virus and T-bet/GFP-expressing virus (T-bet); or (4) Chop/CD90.1-expressing virus and T-bet/GFP-expressing virus (Chop/T-bet). Cells were primed for 24 h and then transduced for additional 72 h in the presence of stimulating antibodies plus IL-2 (50 U/ml). Then IFNγ levels were detected by FACS in gated CD90.1+GFP+ cells. Right: Representative FACS result; left: Merged results from three independent experiments. In the bar graphs showing mean ± s.e.m., *p < 0.05, **p < 0.01 were calculated using two-tailed unpaired Student's t test

interaction of Chop with additional major mediators of CD8+ T cell function remain unknown. Initial research suggests the potential interaction between Chop and key mediators of T cell function, including NFkb, NFAT, and AP1[51]. This suggests a key role of Chop in the overall function of T cells.

Although our results focused on understanding the role of Chop in CD8+ T cells, previous reports have suggested a potential effect of Chop in the function of CD4+ T cells. Chop overexpression or activation of amino acid starvation signals blunted the development of Th17 populations[50,59]. Also, deletion of the ER chaperone gp96 promoted the development of undifferentiated CD62Lhigh/CD44low CD4+ T cells that exhibited enhanced antitumor activity[36]. In line with those reports, our unpublished data show a higher Th1 frequency in activated *Ddit3* null CD4+ T cells. In addition, our recent report pointed on the role of the activation of Xbp1 in the dysfunction of CD4+ T cells occurring in ovarian cancer[60]. Despite the potential relevance of these findings, the role of Chop in the regulation of CD4+ T cell function in tumors remains to be tested.

In summary, our results demonstrate the primary role of Chop in the impaired activity CD8+ T cells in tumors and suggest the feasibility of overcoming tumor-induced CD8+ T cell suppression and increasing the efficacy of T cell-based immunotherapy by blocking Chop or ER stress.

## Methods

**Mice and cell lines.** Experiments using mice were developed through an approved Institutional Animal Care and Use Committee (IACUC) protocol (IS00004043) and an active Institutional Biosafety Committee (IBC) study (#1385), both reviewed by the Integrity and Compliance board at the University of South Florida and Moffitt Cancer Center. Thus, the presented work has complied with all the relevant ethical regulations for animal testing and research. Wild-type C57BL/6 mice (6–8 weeks) were purchased from Envigo. *Ddit3*flox/flox mice were a gift from Dr. Tabas, Columbia University (currently available from the Jackson Laboratory as B6.Cg-*Ddit3*tm1.1Irt/J). *Ddit3* deficient (−/−) (B6.129S (Cg)-*Ddit3*tm2.1Dron/J, #005530), *Eif2ak3*flox (Eif2ak3tm1.2Drc/J, #023066), *CD4*-cre (Tg(Cd4-cre)1Cwi/BfluJ, #022071), Pmel (B6.Cg-Thy1a/Cy-Tg (TcraTcrb)8Rest/J, #005023), and OT-1 (C57BL/6-Tg(TcraTcrb) 1100Mjb/J, #003831) mice were obtained from the Jackson Laboratory. *Ddit3*T cell-KO mice were created by crossing *Ddit3*flox and *CD4*-cre mice. Animal colonies were maintained in the comparative medicine facilities at Moffitt Cancer Center. Tumor cell lines were obtained from American Type Culture Collection (ATCC) or Kerafast. All cell lines were validated to be mycoplasma-free using the Universal Mycoplasma Detection Kit (#30-1012K, ATCC). For tumor growth, 3LL lung carcinoma (#CRL-1642, ATCC), B16-F10 melanoma (#CRL-6475, ATCC), EL-4 thymoma (#TIB-39, ATCC), and MCA-38 colon carcinoma cells (#ENH204, Kerafast) were injected s.c. into the mice and tumor volume was calculated using the formula: (small diameter)² × (large diameter) × 0.5. For developing in vivo ovarian ID8-*Defb29*/*Vegf-a* tumors, ID8-*Defb29*/*Vegf-a* tumor cells (provided by Dr. Conejo-Garcia[31,32]) were injected

intraperitoneally (i.p.) into syngeneic C57BL/6 mice and body weight evaluated daily. Mice with a weight gain >30% were sacrificed and the ascites collected. To deplete CD8+ T cells, tumor-bearing mice were injected i.p. with 400 μg α-CD8 antibody (Lyt 2.1, #BE0118, BioXcell) at day 0, followed by every third day until tumor endpoint.

**Patient population.** Tumor suspensions and blood from de-identified patients with advanced ovarian carcinoma and blood from healthy donors were obtained from a tissue repository established by Dr. Conejo-Garcia (Moffitt Cancer Center). In addition, T cells were isolated from de-identified buffy coats from healthy blood donors (One-Blood). All human studies were covered through the approved Institutional Review Board (IRB) exempt protocol #19223, previously reviewed by the Regulatory Affairs Committee Board at Moffitt Cancer Center. All de-identified patients signed approved consent forms. Also, an ovarian tissue micro-array (TMA; from the Moffitt Cancer Center Biorepository) and clinical data were available for 87 de-identified pathologically confirmed high-grade advanced serous epithelial ovarian carcinomas and 12 healthy ovary and fallopian tube tissues. Clinical information was analyzed in coordination with the leading pathologist (Dr. A. Magliocco) and the Biostatistics Service Core at Moffitt.

**Plasmids and viral vectors.** Retro-Chop expressing vector, pMSCV-*Ddit3*-IRES-Thy1.1, was created by inserting the full-length *Ddit3* cDNA sequence into the retroviral vector MSCV-IRES-Thy1.1 (between BglII and HpaI sites). The backbone vector was a gift from Dr. Zhichun Ding (Augusta University). Retro-vector pBMN-8x-CRE-GFP coding for the 8x-Chop response elements (8x-CRE) was created after cloning the 300 bp edited *Tbx21* promoter region that contains 8 discontinuous CRE repeats into the pBMN-IRES-GFP vector (#1736, Addgene, between BamHI and NotI sites). Retro-vector pBMN-T-bet-GFP coding for T-bet was created via inserting the full-length *Tbx21* cDNA sequence into the pBMN-IRES-GFP vector (between BamHI and NotI sites). For retrovirus production, 293T cells (#CRL-3216, ATCC) were co-transfected with the retroviral expression vector and the packaging vector that expresses the ecotropic envelope, and viral supernatants were collected 48–72 h post-infection. *Tbx21*-CRE luciferase plasmid was created by cloning a sequence containing two CRE on the *Tbx21* promoter (2x-CRE) into the SV40 promoter driving Firefly luciferase, followed by cloning into pGL3 basic Luciferase Reporter Vector (#E1751, Promega).

**Antibodies.** The following antibodies were used for flow cytometric studies: mouse IgG1 isotype (1:100, #GM4992), mouse IgG2a kappa isotype (1:200, #12-4724-82), mouse IgG1 kappa isotype (1:200, #17-4714-42), CD3e (1:200, 145-2C11, #11-0031-82), CD4 (1:200, GK1.5, #12-0041-82), CD8a (1:200, SK1, #17-0087-42), CD8a (1:200, HIT8a, #11-0089-42), CD44 (1:200, IM7, #17-0441-82), CD62L (1:200, MEL-14, #12-0621-82), CD90.1 (1:200, HIS51, #11-0900-81), CD103 (1:200, B-Ly7, #17-1038-42), IFN gamma (1:200, XMG1.2, #11-7311-82), and TNF alpha (1:200, MP6-XT22, #11-7321-82) antibodies were purchased from eBioscience. Mouse IgG2a kappa isotype (1:200, MOPC-173, #400260 and #400273), CD45 (1:200, 30-F11, #103112), T-bet(1:100, 4B10, #644814), and KLRG1 (1:200, 2F1, #138411) antibodies were obtained from Biolegend. CD69 (1:200, H1.2F3, #553237) and CD127 (1:200, SB/199, #562959) antibodies were from BD Biosciences. The following antibodies were used for immunoblotting: Phospho-Perk (T980) (1:1000, 16F8 rabbit polyclonal, #3179), Perk (1:1000, C33E10 rabbit polyclonal, #3192), IREα (1:1000, 14C10 rabbit polyclonal, #3294), Atf4 (1:1000, D4B8 rabbit polyclonal, #11815), GCN2 (1:1000, rabbit polyclonal, #3302), Granzyme B (1:2000, rabbit polyclonal, #4275), and EOMES (1:1000,

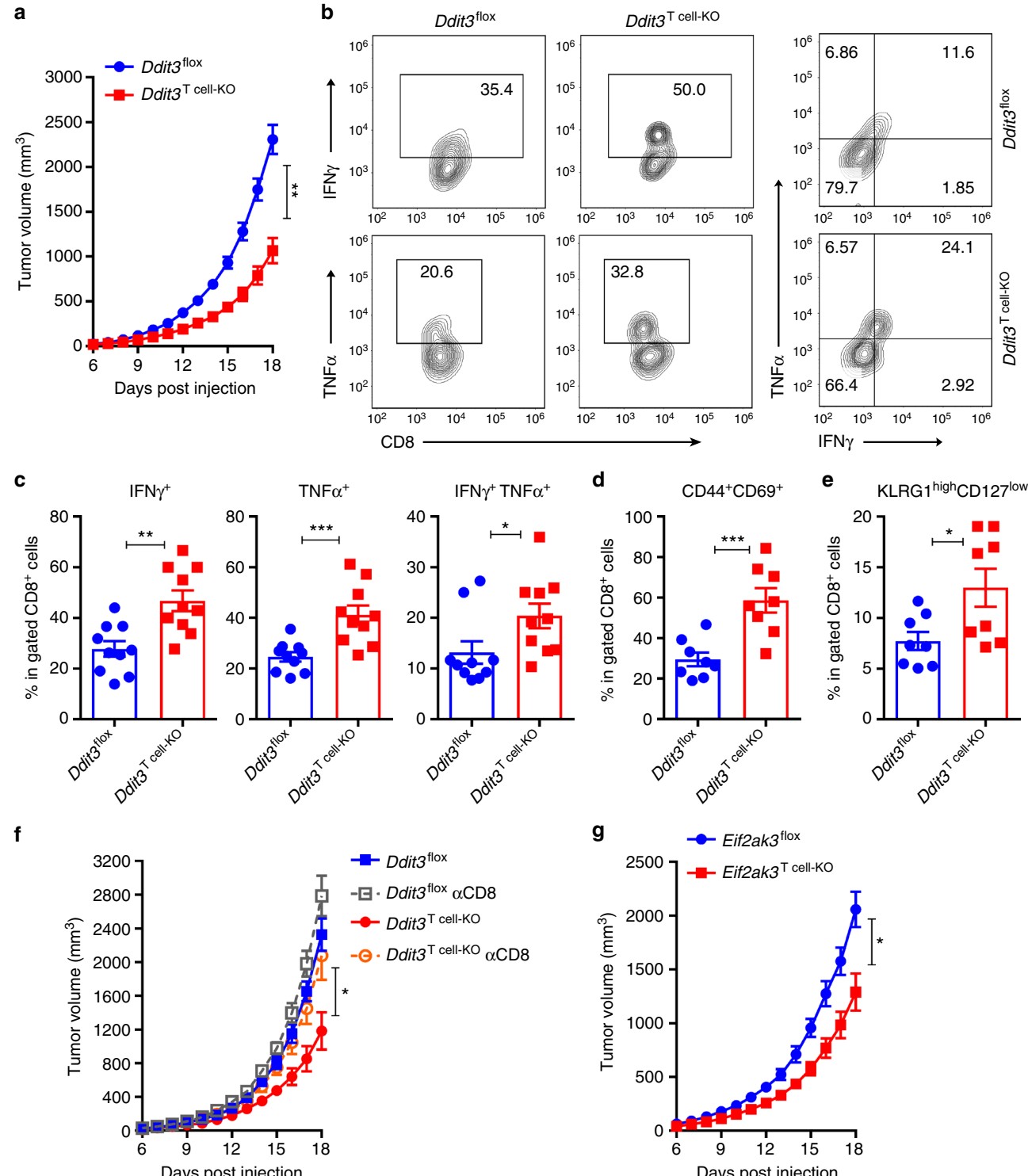

**Fig. 5** *Ddit3* null T cells exert promoted protective antitumor immunity. **a** Tumor growth in *Ddit3*flox (blue) or *Ddit3*T cell-KO (red) mice bearing B16 tumors. Average kinetics ± s.e.m (*n* = 15 mice/group). **b** Left: Representative result of IFNγ+ (upper panel) and TNF-α+ (lower panel), and right: IFNγ+ TNF-α+ in gated CD45+ CD8+ TILs from *Ddit3*flox or *Ddit3*T cell-KO mice bearing B16 tumors. Tumors were collected 18 days after B16 tumor injection. (*n* = 10 tumors/group). **c** Percentage of IFNγ+ (left), TNF-α+ (middle) and IFNγ+ TNF-α+ (right) CD8+ tumor-infiltrating T lymphocytes (TILs) in B16 tumors isolated from *Ddit3*flox or *Ddit3*T cell-KO mice. (*n* = 10 mice/group). **d** Frequency of CD44+ CD69+ effector CD8+ TILs in B16 tumors isolated from *Ddit3*flox or *Ddit3*T cell-KO mice. (*n* = 8 tumors/group). **e** Percentage of KLGR1high CD127low late effector CD8+ TILs in B16 tumors isolated from *Ddit3*flox or *Ddit3*T cell-KO mice. (*n* = 8 tumor/group). **f** B16 tumor growth in *Ddit3*flox (blue) or *Ddit3*T cell-KO (red) mice, with or without 400 μg α-CD8 antibody injection on every third day (green or orange, respectively). Average tumor volume kinetics ± s.e.m in 6 mice/group. **g** Tumor growth in *Eif2ak3*flox (blue) or *Eif2ak3*T cell-KO (red) mice bearing B16 tumors. Average kinetics of tumor volume ± s.e.m in 8 mice/group. In the bar graphs showing mean ± s.e.m., *p < 0.05, **p < 0.01, ***p < 0.001 were calculated using two-tailed unpaired Student's *t* test

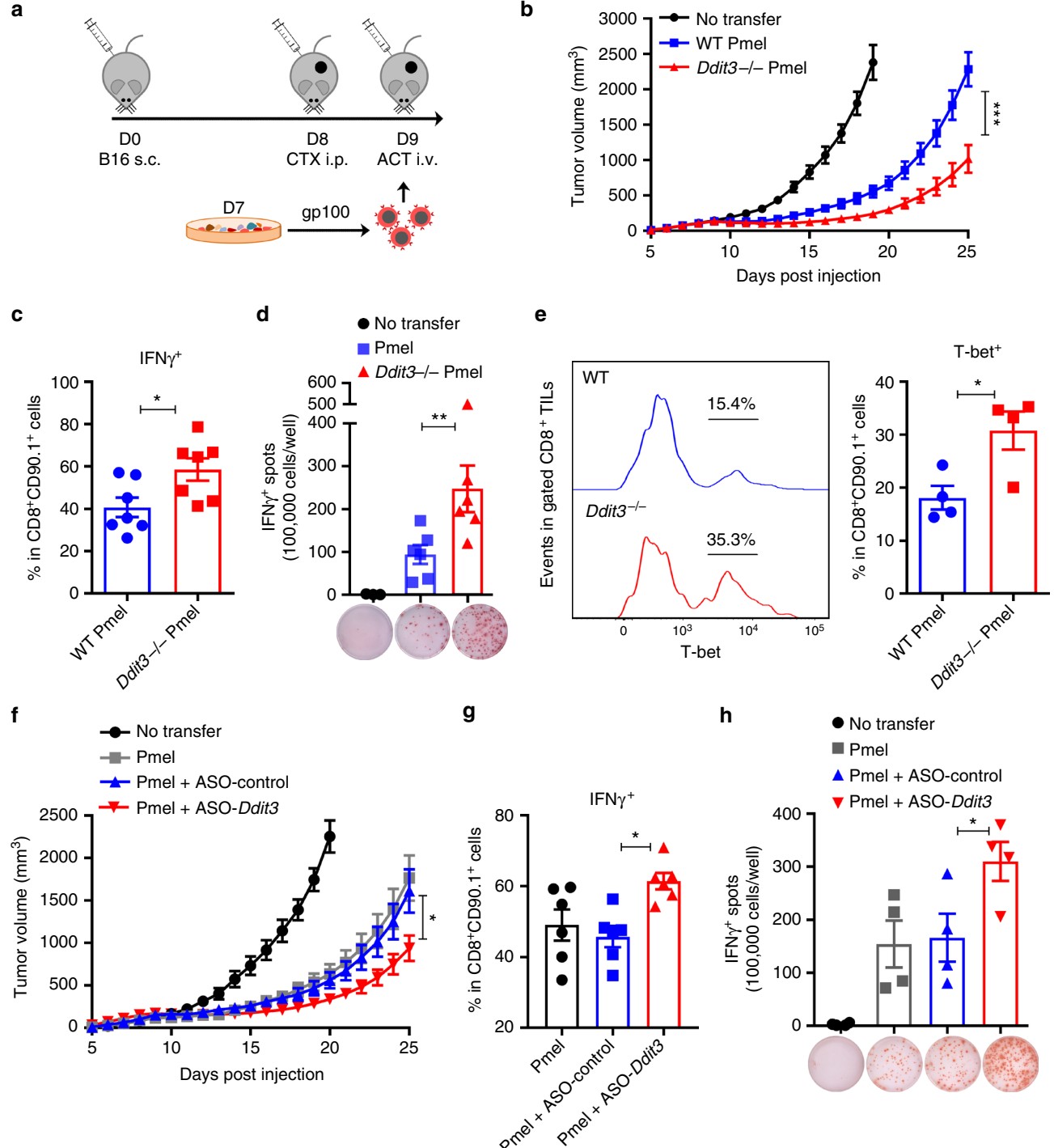

**Fig. 6** *Ddit3* deletion or silencing increases the effect of T cell-based immunotherapy. **a** Working model of Pmel T cell adoptive transfer. CD90.2+ wild-type mice were injected subcutaneous with B16 tumors, and after 8 days, they received a single dose of cyclophosphamide (CTX). The next day, mice received 1 × 10[6] primed wild-type or *Ddit3*−/− or anti-sense oligonucleotide (ASO)-treated CD90.1+CD8+ Pmel T cells. **b** Tumor growth in B16-bearing mice (black) transferred with wild-type (blue) or *Ddit3*−/− (red) CD90.1+ CD8+ Pmel T cells. Average kinetics of tumor volume ± s.e.m. in 9 mice/group. **c** Percentage of IFNγ+ cells in CD90.1+ CD8+ tumor-infiltrating T lymphocytes (TILs) from B16 tumors 5 days after wild-type or *Ddit3*−/− CD8+ Pmel T cell transfer. (*n* = 7 tumors/group). **d** ELISpot for interferon-γ (IFNγ) in the spleens of B16 tumor-bearing mice 5 days after wild-type or *Ddit3*−/− Pmel T cell transfer exposed to gp100$_{25-33}$ for 24 h. (*n* = 6 spleens/group). **e** T-bet in CD90.1+CD8+ TILs from B16 tumors 5 days after wild-type or *Ddit3*−/− CD8+ Pmel T cell transfer. Right: representative result; left: merged results from n = 4/group. **f** Tumor growth in B16-bearing CD90.2+ mice (black) that received wild-type CD90.1+CD8+ Pmel T cells that were non-treated (gray) or pretreated with ASO-Control (blue) or ASO-*Ddit3* (red). Average tumor volume kinetics ± s.e.m. of 8 mice/group. **g** Percentage of IFNγ+ in CD90.1+ CD8+ TILs from B16 tumor-bearing mice transferred with control, ASO-Control, or ASO-*Ddit3*-pretreated CD90.1+ Pmel T cells. (*n* = 6 tumors/group). **h** IFNγ ELISpot in spleens collected from B16 tumor-bearing mice treated with Pmel T cells or ASO-Control or ASO-*Ddit3*-pretreated Pmel T cells. Spleens were collected 5 days after T cell transfer and exposed 24 h to gp100$_{25-33}$. (*n* = 4). Bar graphs show mean value ± s.e.m. and *$p < 0.05$, ***$p < 0.001$ were calculated using two-tailed unpaired Student's *t* test

D8D1R rabbit polyclonal, #81493) antibodies were purchased from Cell Signaling Technology. CHOP (1:500, 9C8, #ab11419), phospho-IREα (S724) (1:1000, rabbit polyclonal, #ab48187), PKR (1:1000, EPR19374 rabbit monoclonal, #ab184257), and ID2 (1:1000, 2457C5a, #Ab53545) antibodies were from Abcam. Additional antibodies included CHOP (1:500, R-20 rabbit polyclonal, #sc-793 from Santa Cruz), T-bet (1:500, 04-46, #37776 from BD Biosciences), HRI (1:1000, 7H3L3, #702551 from Invitrogen), BLIMP1/PRDM1 (1:1000, 3H2-E8, #NB600-235SS from Novus Biologicals), and GAPDH (1:2000, 6C5, #10R-G109A from Fitzgerald).

**Antibody labeling**. Anti-Chop (Anti-DDIT3 antibody, 9C8, #ab11419) and the isotype control antibody Mouse IgG2b (PLPV219, #ab91366) were purchased from Abcam. Anti-CHOP and isotype antibody labeling was performed using the Site-Click™ R-PE Antibody Labeling Kit (#S10467, Thermo-Fisher), following the manufacturer's instructions.

**Western blot**. Cell lysates were electrophoresed in 4–15% Mini-PROTEAN® TGX™ Precast Protein Gels (Bio-Rad), transferred to PVDF membranes by iBlot™ Gel Transfer Device (ThermoFisher), and blotted with the corresponding primary and secondary antibodies. Membrane-bound immune complexes were detected by ChemiDoc™ Imaging System (Bio-Rad). Densitometric analyses were completed using the Image Lab™ Software (Bio Rad). Uncropped and unprocessed scans of the most important immunoblot results are presented in Source Data files.

**T cell isolation, activation, and transduction**. Mouse CD3$^+$, CD4$^+$, and CD8$^+$ T cells were isolated from the spleens using negative enrichment kits (EasySep, Stem Cell Technology Kits) and activated with plate-bound 0.5 μg/ml anti-CD3 (145-2C11, #553058, BD Biosciences) plus CD28 (37.51, #553294, BD Biosciences). Stimulation of Pmel or OT-1 T cells was developed after activation of splenocytes with 1 μg/ml gp100$_{25–33}$ (KVPRNQDWL, #AS-62589, Anaspec) or OVA$_{257–264}$ (SIINFEKL, AS-60193-5, Anaspec), respectively, and CD8$^+$ T cells enriched at endpoints by negative selection kits (Stem Cell technologies). Human CD3$^+$ T cells were isolated from human peripheral blood mononuclear cell using negative enrichment kits (Life Technologies) and activated with soluble anti-CD3 (1 μg/ml, OKT3, #16-0037-85, ThermoFisher) and anti-CD28 (0.5 μg/ml, L293, #340975, BD Biosciences) antibodies in 24-well plates bound with 10 μg/ml goat anti-mouse (GAM, #01-18-06, KPL). For transduction, retroviruses were obtained from 293T cells co-transfected with the retroviral expression vector and the packaging vector that expresses the ecotropic envelope. For retroviral transduction, T cells were isolated from the spleens and stimulated for 48 h with Dynabeads magnetic beads (bead-to-cell ratio 1:1, #11-456-D, ThermoFisher Scientific), after which T cells and retroviruses were mixed in 24-well plates pre-coated with 25 μg/ml RetroNectin (Recombinant Human Fibronectin Fragment, #T100B, Takara), following the vendor's instructions. For siRNA transfection, freshly isolated splenic CD8$^+$ T cells were primed with plate-bound anti-CD3/CD28 and mixed with a mock or a pool of four gene-specific siRNA (Dharmacon™ Accell™ siRNA, GE Healthcare), following the suggested Accell™ delivery protocol. Accell siRNAs were the following: Non-targeting Control siRNA (#D-001950-01-05), Atf4 (Gene ID: 11911, #E-042737-00-0005), Eif2ak1 (Gene ID: 15467, #E-045523-00-0005), Eif2ak2 (Gene ID: 19106, #E-040807-00-0005), Eif2ak3 (Gene ID: 13666, #E-044901-00-0005), and Eif2ak4 (Gene ID: 27103, #E-044353-00-0005).

**Tumor digestion and TIL fluorescence-activated cell sorter (FACS) sorting**. Tumors were digested with DNASe I and Liberase (Roche USA, Branchburg, NJ). Tumor digests were then treated with ACK buffer (Ammonium-Chloride-Potassium) to lyse the red blood cells. For CD8$^+$ T cell sorting, tumors were labeled with Zombie Violet™ Fixable Viability probe, antibodies against CD45 and CD8 and further sorted via a FACS-Aria II.

**FACS staining**. For surface staining, cells were labeled with the appropriate antibodies in the presence of Fc blocker. For intracellular staining, surface-labeled cells were fixed with Cytofix/Cytoperm™ Solution (BD Biosciences), washed in Permwash™ 1× solution (BD Biosciences), and labeled with intracellular antibodies. Cells were then washed in Permwash™ 1× and phosphate-buffered saline (PBS) and acquired by FACS. For the staining of Chop and T-bet, T cells were fixed with the Transcription Factor Fixation and Permeabilization Kit (eBioscience), using the recommended protocol from the vendor. Cell live vs. dead discrimination was performed prior to antibody labeling by Zombie Violet™ Fixable Viability probe (BioLegend). FACS acquisition was performed in a CytoFLEX II, Beckman Coulter.

**Adoptive cell transfer therapy**. CD90.2$^+$ C57BL/6 mice were s.c. injected with B16 cells, and 8 days post-tumor injection, they received a single i.p. injection with 100 mg/kg cyclophosphamide. The next day, mice were adoptively transferred intravenous (i.v.) with 1 × 10$^6$ pre-primed wild-type or Ddit3−/− or ASO-Ddit3 or ASO-Control CD90.1$^+$ Pmel T cells. For ASO treatments, T cells were treated with 10 μM Ddit3-specific LNA-Gapmer (ASO-Ddit3, Sequence: +C*+T* +A*G*C*T*G*T*G*C*C*A*C*+T*+T*+T. (+): LNA base, (*): Phosphorothioate linkage) or control LNA-Gapmer (ASO-Control, Sequence: +C*+G* +T*T*T*A*G*G*C*T*A*T*G*T*A*+C*+T*+T) from Secarna Pharmaceuticals.

ELISpot and FACS assays were performed on day 5 post-transfer. For vaccination, 1 × 10$^6$ naive carboxyfluorescein succinimidyl ester (CFSE)-labeled wild-type Pmel CD8$^+$ T cells were adoptively transferred i.v. into C57BL/6 mice. The same day, 5 μg gp100$_{25–33}$ in incomplete Freund's Adjuvants (IFA) were injected s.c. and lymph nodes were collected 96 h later to monitor Chop expression in proliferating CD90.1$^+$ Pmel T cells by FACS.

**IFNγ ELISpot**. Five days after wild-type or Ddit3−/− Pmel CD8$^+$ T cell transfer, the spleens from B16-bearing mice were harvested and 1 × 10$^5$ cells splenic cells were seeded in ELISpot plates containing pre-bound IFNγ capturing antibody in the presence or absence of 1 μg/ml gp100$_{25–33}$. Production of IFNγ was detected 24 h later by ELISpot using the Mouse IFNγ ELISpot ELISA Ready-SET-Go™ Kit (#88-7384, eBioscience).

**ChIP, DNA-binding ELISA, and luciferase gene reporter assay**. ChIP assays were performed using the SimpleChIP Enzymatic Chromatin IP Kit (Cell Signaling Technology). Chromatin was prepared from 4 × 10$^6$ wild-type or Ddit3−/− cells that were activated for 72 h. Sheared chromatin was immunoprecipitated with antibodies against Chop, Histone 3 (D2B12 Rabbit antibody, #4260, Cell Signaling Technology), or rabbit IgG isotype (DA1E, #39600, Cell Signaling Technology). Purified DNA was analyzed by quantitative PCR with pre-validated primers that detect the Chop-binding sequence in the Tbx21 promoter (#GPM1043970(-)01A, SABiosciences-Qiagen). For DNA-binding activity, nucleic protein extracts were isolated from activated control and Ddit3 null T cells and monitored for binding of T-bet to a consensus DNA sequence through a specific TransAM® Kit (#51396, Active-Motif). For the dual luciferase gene reporter assays, plasmids harboring the Tbx21-CRE-SV-40 firefly luciferase and the SV-40 promoter-driven Renilla luciferase were co-transfected into 293T cells with pMSCV-Ddit3-IRES-Thy1.1 or control pMSCV- IRES-Thy1.1 vectors. After 48 h, dual luciferase assay was performed by Dual-Luciferase® Reporter Assay System (#E1910, Promega).

**Quantitative PCR**. Total RNA was isolated from T cells using TRIzol (Life Technologies). Reverse transcription was performed using the Verso cDNA Synthesis Kit (Thermo Scientific). Quantitative PCR reactions were prepared by using Bio-Rad SYBR green master mix and performed on an Applied Biosystems thermocycler (7900 HT). Primers against murine Ddit3 forward (GGAGCTG-GAAGCCTGGTATG) reverse (GGATGTGCGTGTGACCTCTG), Atf4 forward (GCCTGACTCTGCTGCTTA) reverse (GCCTTACGGACCTCTTC), Eif2ak3 forward (ATCGCAGAGGCAGTGGAGTT) reverse (AGGCTGGCATTGGAGT-CAGT), and Actb forward (TGTGATGGTGGGAATGGGTCAGAA) reverse (TGTGGTGCCAGATCTTCTCCATGT) were from IDT. Primers for murine Il12b2, Cxcr3, Ifng, Gzmb, Tbx21, Cbfa3, Eif2ak1, Eif2ak2, and Eif2ak4 and human DDIT3 were purchased from QIAGEN. Relative expression was calculated using the ΔΔCt method and normalized to actb levels.

**RNA-Seq analysis**. Total RNA from activated wild-type and Ddit3−/− Pmel CD8$^+$ T cells was obtained using the RNAeasy Mini Kit (Qiagen). RNA was quantified in a NanoDrop 1000 and RNA quality assessed by Agilent 2100 Bioanalyzer. Samples were then processed for RNA-sequencing using the NuGen Ovation Mouse RNA-Seq Multiplex System (NuGEN Technologies). Briefly, 100 ng of RNA were used to generate cDNA and a strand-specific library following the manufacturer's protocol. Quality control steps including BioAnalyzer library assessment and quantitative PCR for library quantification were performed. The libraries were then sequenced in an Illumina NextSeq 500 v2 sequencer with 75-base single-end run in order to generate approximately 60 million reads per sample. Sequencing reads were subjected to adaptor trimming, series of quality assessment before being aligned with Tophat v2.0.13 against mouse reference genome mm10. Quantification of read counts aligned to the region associated with each gene was performed using HTSeq v0.6.1 based on RefSeq gene model. Read counts of all samples were normalized using the median-of-ratios method implemented in R/Bioconductor package DESeq2 v1.6.3. Differential expression analysis between wild-type and Ddit3−/− Pmel CD8$^+$ T cells was performed by serial dispersion estimation and statistical model fitting procedures implemented in DESeq2. Genes with a p value adjusted for multiple testing with the Benjamini–Hochberg correction of <0.05 were determined to be significantly differentially expressed. GSEA was used to assess significant enrichment of specific biological pathways or gene sets. GSEA was performed with a list of pre-ranked genes based on their differential expression between the two groups using the GSEA pre-ranked function in the Gsea-3.0 software against C7 immunological signatures in the Molecular Signatures Database (MSigDB) v6.1.

**Seahorse analyses**. OCR and ECAR were measured using a XF96 extracellular flux analyzer (Seahorse Bioscience). Purified wild-type or Ddit3−/− CD8$^+$ T cells were activated with anti-CD3/CD28 for 72 h. Then, T cells were collected, washed, plated onto CellTak pre-coated wells (1 × 10$^5$ T cells/well), and subjected to Seahorse XF Cell Mito Stress Test or Seahorse XF Glycolysis Stress Test (both from Agilent) using specific non-buffered XF base mediums. For the Mito Stress analysis, T cells were analyzed under basal conditions and in response to 1 μM

oligomycin, 1 µM fluorocarbonyl-cyanide-phenylhydrazone, and 0.5 µM Rotenone/Antimycin A (all from Sigma-Aldrich). For the Glycolysis Stress Test, T cells were plated in XF media lacking glucose and monitored under basal conditions and in response to 10 mM glucose, 1 µM oligomycin, and 50 mM 2-deoxy-D-glucose (all from Sigma-Aldrich).

**In vitro cytotoxic assays**. Wild-type and $Ddit3-/-$ Pmel cells were stimulated with 1 µg/ml gp100$_{25-33}$ for 48 h, after which CD8$^+$ T cells were co-cultured with a mixture of EL-4 cells (50% gp100$_{25-33}$-loaded EL-4 cells labeled with 1 µM CFSE and 50% control peptide-loaded EL-4 cells labeled with 0.1 µM CFSE). Proportion of different sets of EL-4 cells was tested 24 h later via FACS.

**Immunofluorescence and imaging analysis**. Two core tissue biopsies per case (2 mm in diameter) were taken from individual paraffin-embedded sections and arranged randomly in the block. The TMA was cut into 4-µm sections and placed on super frost charged glass slides. After de-paraffinization and antigen retrieval, sections were blocked in 5% goat serum and incubated overnight with mouse monoclonal anti-CD8 (1:100, IgG1, C8/144B, #108M-98, Cell Marque) and mouse monoclonal anti-Chop (1:100, IgG2b, 9C8, #ab11419, Abcam) or mouse monoclonal anti-CD8 and rabbit polyclonal anti-CHOP/GADD153 (1:500, R-20, #sc-793, Santa Cruz Biotechnologies), followed by washes in PBS and incubation in secondary goat anti-mouse IgG1 and IgG2b or goat anti-mouse IgG1 and anti-rabbit IgG labeled with Alexa Fluor® 488 and 647 (all 1:200, ThermoFisher Scientific), respectively. For isotype control staining, tissues were similarly incubated with mouse IgG1 isotype (1:100, MG1-45, #401402, Biolegend) and mouse IgG2b isotype (1:100, MPC-11, #400376, Biolegend) or mouse IgG1 (1:100) and rabbit IgG (1:500, DA1E, #39600, Cell Signaling Technology), followed by incubation with specific secondary antibodies labeled with Alexa Fluor® 488 and 647, respectively. Next, slides were washed in PBS, mounted in aqueous mounting media with 4,6-diamidino-2-phenylindole (DAPI; Thermo-Fisher). Expression of Chop was compared among the CD8$^+$ cells in both groups, and representative images were taken in a Leica SP8 Confocal microscope. TMA histology slides were scanned using the APERIO ScanScope FL (Leica Biosystems, Wetzlar, Germany). Briefly, high-resolution (×20) monochromatic 8-bit digital images were obtained for each TMA core using separate filters to define the nuclear (DAPI), CD8 (fluorescein isothiocyanate (FITC)), and CHOP (Cy5). Images were stored in Aperio's Spectrum software. The TMA image was segmented into individual cores using the software's TMA lab module. The individual core images were imported into the Definiens Tissue Studio v4.2 suite cellular analysis (Definiens Inc, Germany). In Tissue Studio, DAPI and FITC stain channels were used to segment each cell nucleus and cytoplasm, respectively, based on intensity and size constraints. The cytoplasm simulation intensity threshold was set to a value where cell boundaries were determined by the difference in auto-fluorescence between cell bodies and background. Intensity values for the FITC and Cy5 channels for each cell were then analyzed.

**Statistical analysis**. Statistical analyses were determined using the Graphpad Software 7 and SAS 9.4. At the 95th percentile distribution, CHOP fluorescence in CD8$^+$ T cells from normal tissues had a value of 15 mean fluorescence intensity (MFI). Thus, CHOP$^+$ cells were categorized as those having >15 MFI. Percentage of CHOP$^+$ cells in the overall CD8$^+$ population in the control and tumor samples was then created and compared using $t$ test. Also, the percentage of CD8$^+$ CHOP$^+$ cells was crossed with the results of ovarian cancer cytoreductive surgery using $t$ test. Furthermore, we used the median of the proportion of CD8$^+$ CHOP$^+$ cells in the tumor (0.41 for TMA stained with Clone 9C8 and 0.37 for TMA stained with Clone R20) to establish the cutoff of CHOP$^{high}$ and CHOP$^{low}$. Therefore, samples considered CHOP$^{high}$ had ≥41% (Clone 9C8) or ≥37% (Clone R20) of CD8$^+$ CHOP$^+$ cells, whereas CHOP$^{low}$ had <41% or <37%, respectively. CD8$^+$ CHOP$^{high}$ and CHOP$^{low}$ samples were compared for overall survival of the patients using Gehan–Breslow–Wilcoxon test. The nominal alpha level for these comparisons was 0.05. Two-tailed unpaired Student's $t$ test was used for most of the statistical analyses and a $p$ value of <0.05 was considered statistically significant. The specific statistical test results are indicated in each figure: *$p < 0.05$; **$p < 0.01$; ***$p < 0.001$.

**Reporting Summary**. Further information on experimental design is available in the Nature Research Reporting Summary linked to this article.

## Data availability

The RNA-Seq raw results that support Fig. 3 of the study have been deposited in the Gene Expression Omnibus database under the accession number GSE112823. The authors declare that all results supporting the findings of this study are available within the paper and its Supplementary Figures. Also, additional supporting data of this study are available in Source Data Files, as well as from the authors upon reasonable request. A reporting summary for this article is available as a Supplementary Information file.

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

## Acknowledgements

The authors would like to thank J. Kroeger from the Flow Cytometry core; S. McCarthy and N. Lopez-Blanco from the CLIA Tissue Imaging core; S. Yoder from the Molecular Genomics Core; Y. Xu from the Cancer Informatics Core; and J. Johnson from the Analytic Microscopy Core. All used cores are partially funded through the NCI designated Moffitt Cancer Center Support Grant (CCSG) P30-CA076292. Support for the used shared resources was provided by the Cancer Center Support Grant (CCSG) CA076292 to Moffitt Cancer Center. This study was supported in part by the National Institutes of Health (NIH) grants: R01CA184185 to P.C.R; R01CA103320 and R01CA211229 to D.H.M.; and R01CA157664, R01CA124515, R01CA178687, and U01CA232758 to J.C.-G. J.R.C.-R. was supported by the Ovarian Cancer Academy-Early-Career Investigator Award W81XWH-16-1-0438 of the Department of Defense (DOD).

## Author contribution

Y.C. and P.C.R. conceived the main ideas and supervised the overall progression of the study; Y.C., J.T.-T., R.A.S., C.A., W.D., E.M., L.C., T.L.C., A.M., D.M., R.K., S.M., F.J., and R.R.R. developed major methodologies, databases, reagents, and primary experiments; Y.C., J.T.-T., E.M., L.C., R.R.R., and C.A. analyzed different aspects of the results; Y.C., A.M., S.M., J.R.C.-R., D.M., J.C.-G., and P.C.R. contributed toward major concepts, reagents, animals, and human samples; Y.C. and P.C.R. wrote the manuscript. All authors discussed the results and edited and commented on the manuscript.

## Additional information

**Competing interests:** The authors declare no competing interests.

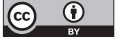

