## [Peer Review File · Nature Communications]

Reviewers' Comments:

Reviewer #1:

Remarks to the Author:

In this manuscript from Rodriguez and colleagues, the authors examined the impact of ER stress on CD8⁺ T cell function in the context of tumor infiltration and control. The authors observed that Chop (C/EBP homologous protein or Ddit3) expression was significantly enhanced in CD8⁺ TILs from various murine tumor models but not in matched spleen controls. Increased CHOP expression was also found in CD8⁺ TILs from humans with ovarian carcinoma. Moreover, increased CHOP nuclear localization was correlated with poor clinical response to cytoreductive surgery and reduced overall survival. In vitro and in vivo activation of wild type and TCR transgenic T cells similarly resulted in increased CHOP expression, which was blocked following treatment with the ER stress attenuator – TUDCA. Activating Perk phosphorylation (Eif2ak3), an upstream regulator of Chop expression, was elevated in stimulated cells. In addition, Chop protein abundance was reduced in activated Eif2ak3-KO T cells indicating that Perk pathway activity is increased in effector cells. Lastly, T cells stimulated in the presence of ascites from mice bearing ovarian tumors or from patients with ovarian cancer displayed augmented Chop expression. Ascites-augmented Chop expression was reduced in the presence of the ROS-scavenger, L-NAC. Activation of Ddit3^{-/-} CD8⁺ T cells revealed key alterations in the expression of metabolic (glycolysis and OXPHOS) and effector genes. IFN- γ and granzyme B expression and cell proliferation were increased in KO T cells. The majority of Ddit3 KO T cells also displayed an effector memory like phenotype CD44⁺CD62L⁻). Ddit3 was found to be a direct repressor of T-bet (Tbx21) gene transcription. Transduction of Tbx21 in Chop overexpressing cells rescued IFN- γ production. Adoptive transfer of Ddit3 KO effector cells into B16 tumor bearing mice delayed tumor growth, moreover, Ddit3 KO TILs demonstrated increased T-bet expression greater IFN- γ and TNF production following ex vivo re-stimulation. In all, these findings demonstrated that the PERK-CHOP pathway is an important negative regulator of cytotoxic T cell function by inhibiting T-bet expression.

The role of cellular metabolism in the activation, differentiation, and expansion of immune cells is well established. Much of what is known regarding CD8⁺ T cell metabolism is often linked to glycolysis and mitochondrial respiration. However, the role of the endoplasmic reticulum (ER), which is a major metabolic organelle in cells, in CD8⁺ T cells is often overlooked. The findings reported in this manuscript point to a new role for the ER in the regulation of T cell effector function and maturation. In particular, linking ER stress-induction by the tumor microenvironment (TME) to TIL dysfunction offers new insight regarding the impact of extrinsic factors (or lack thereof) on TIL elimination of tumor cells. A newly published article from the Cubillos-Ruiz laboratory (Minkyung, Nature 2018) have also observed a similar role for another ER stress pathway, IRE1a-XBP1, on the attenuation of TIL activity in the TME, highlighting the timeliness and importance of the findings reported in this manuscript.

Specific critique:

Figure 1E,F. The authors maintained that CHOP nuclear localization was enhanced in CD8⁺ TILs from patients with ovarian carcinoma. However, the images provided in the Figure do not show nuclear localization, rather CHOP was predominantly found in the cytoplasm. The authors need to show more convincing images (higher magnification?) that demonstrate CHOP nuclear localization.

Figure 2A,C,D,H,I. These immunoblot images lack proper quantification. The authors need to add graph summaries of these subfigures.

Figure 2B. The authors examined Chop expression in proliferating cells, however in the figure legend it was stated that the authors probed for expression in CFSE⁺ cells. The authors should only assess Chop MFI in CFSE low cells.

Figure 2C. The authors need to demonstrate whether PERK phosphorylation was also enhanced following TUDCA or thapsigargin treatment.

Figure 2D. CHOP induction following PERK activation is mediated by the transcription factor ATF4. Is ATF4 protein abundance similarly enhanced in mouse and human T cells following stimulation? The authors need to probe for ATF4 expression using a similar time-course.

Figure 2H. The authors show that addition of ascites from tumor bearing mice reduces PERK pathway activity and Chop expression. However, in this figure Chop induction following activation for 48 h was noticeably lower compared to Fig 2A. The authors need to show a graphic quantification of these experiments to fully appreciate the effect of ascites on PERK pathway activity. See comment relating to Figure 2A,C,D,H,I.

Sup Figure 4C. GSEA of WT and Ddit3^{-/-} T cells revealed that CHOP deletion promoted the expression of OXPHOS and glycolysis genes. Since these two pathways are important for T cell expansion and effector function, the authors need to examine the metabolic signature of primed Ddit3^{-/-} CD8⁺ T cells compared to wild type controls by directly probing for glycolysis and mitochondrial respiration.

Figure 4A. The authors examined the role of Chop in the regulation of T-bet expression. However, in this subfigure T-bet protein abundance at 48 h post-activation is considerably lower in wt versus Ddit3^{-/-} CD8⁺ T cells, which is in stark contrast to previously published findings from Anjana Rao's group and other laboratories (Cruz-Guilloty, JEM, 2009). The authors need to reconcile these data with past findings by performing a time course examination of T-bet protein abundance (similar to Figure 2A for Chop) to highlight the inverse relationship between T-bet and Chop.

Figure 4B,C. The authors demonstrated that the effector maturation program is enhanced in Ddit3 deficient CD8⁺ T cells. However, the authors have not appropriately examined the differentiation status of these cells. Are Ddit3^{-/-} CD8⁺ T cells differentiating into terminally differentiated T cells? It was shown in Fig 3I that these cells display an effector memory phenotype, as evidenced by an enhanced potential to differentiate into CD44⁺CD62L^{neg} cells. However, these are not the appropriate markers to evaluate the differentiation status of newly primed cells. The authors should establish whether differentiation of Chop deficient T cells into effector/memory subpopulations is altered by using KLRG1 and CD127 as surface markers. These two markers should appropriately mark early differentiation into short-lived effector and memory T cells.

Figure 4D-H. The authors elegantly examined the importance of Chop in the repression of T-bet expression. However, in Figure 3B, expression of 3 other transcription factors –Id2, Prdm1 and Eomes– that regulate the differentiation of activated CD8⁺ T cells were also enhanced in Ddit3 deficient cells. The authors need to determine whether protein abundance of all three transcription factors is similarly enhanced as T-bet in Ddit3^{-/-} cells, and whether these proteins are directly regulated by Chop, by analyzing for putative Chop binding sites in their respective promoters using the ChampionChip database.

Figure 5B,C. Polyfunctional T cells have been demonstrated to be important for control of many cancer types (Zhao, E., Nat Imm, 2016). Are the frequencies of polyfunctional (IFN-gamma and TNF double positive) cells also increased in Chop deficient cells?

Figure 5D-G. Mislabeled in text and figure.

Reviewer #2:

Remarks to the Author:

The study from Cao and colleagues study the role of the transcription factor Chop during T cell activation and in particular in CD8 T cells infiltrated in tumors. They showed that Chop was overexpressed in T cells infiltrated in transplantable mouse tumor cell lines as well as in human samples for ovarian cancer. They established that Chop expression was regulated by the ER stress associated kinase Perk, and that Chop was dampening Tbet expression, leading to a decrease effector capacity of the T cells. In a tumor setting, they showed that T cell specific KO mice had a slightly higher capacity to control tumor development. Using TCR transgenic T cells in an adoptive transfer setting, they also showed that the KO T cells have a higher capacity to control tumor growth.

Altogether, the data are convincing and original. I would recommend accepting this article after some modifications, after clarifying a few points and adding additional experiments (for the statistical analysis).

Concerns:

CHOP imaging on ovarian tumor samples from patients shows that CHOP is expressed everywhere in the tumor and not specifically in the CD8 T cells. The authors state that nuclear CHOP level in CD8 correlates with a better survival. Is it not just the general level of CHOP in the entire tumor and not specifically in the CD8 T cells that is correlated with survival? Furthermore the images shown in Figure 1 do not convince me that CHOP is intra-nuclear in the CD8 T cells. I would like to see pictures of what the authors consider as high nuclear content versus a low.

Concerning figure 2, most of the readouts have been carried out with western blots. There are no quantification of the data. Furthermore, the anti-CHOP Ab seems to work properly for flow cytometry according to figure 1. The results obtained by western blots have to be confirmed by flow cytometry for CHOP expression.

In general, for all the figures, I would prefer to see dots and not plain columns for the histograms. This way we can see how many mice or samples have been used. Some statistical analysis have been performed with only 3 replicates per condition. The number of samples should be increased for those. It occurred in several histograms.

For figure 3H and D, only one dot plot is shown for each. Combined results for several experiments have to be shown too.

For figure 4H there is no mention about the number of experiments done.

In the legend for figure 5B, it is mentioned that compiled results for 5 mice are shown... I don't have these results on the figure.

There are some mislabeled figures in the text compared to the figures (Figure 5D instead of 5F and 5E instead of G).

For figure 5F, it is indicated that only 3 mice per group have been used. This number has to be increased. Similarly, experiment for figure 5G has to be repeated (4 mice per group).

In figure 6C the number of mice shown is not indicated. Similar remark for figure 6E.

For figure 6F there are 4 mice per group only. This experiment has to be repeated.

Reviewer #3:

Remarks to the Author:

Severe ER stress-induced mediator C/EBP homologous protein thwarts effector T-cell 1 activity in tumors through repression of T-bet by Yu Cao et al.:

Summary:

. The authors observe that nuclear CHOP is elevated in the splenic and TIL associated CD8CD45 T cells in syngeneic models of cancer. CHOP is elevated in T cells from ovarian cancer patients and negatively correlates with survival nuclear CHOP in suggesting that CHOP expression in CD8+ TILs is a key predictor of poor clinical responses in ovarian cancer.

. CHOP protein is induced at late time points 24-48 hours following costimulation.

. The authors provide genetic evidence that PERK is activated in response to T cell activation and is required for CHOP induction. PERK is required for CHOP induction in TILs. ROS production in T cells

is required for PERK activation.

. Ddit3^{-/-} T cells proliferate faster and have more effector function. Transcripts associated with effector function are upregulated in the absence Ddit3, including metabolic pathways, elevated levels of the cytotoxic mediator granzyme B, and higher IFN γ . Treatment of T cells with treated with the ER stress inhibitor TUDCA, phenocopies the observation of CHOP deficient cells suggesting that by suppressing ER stress pharmacologically, T cells may be more active and demonstrate a greater anti-tumor response.

. Loss of Ddit3 is correlated with upregulation of TBet and TBet target genes Ifng, Il12b2, Cbfa3, and Cxcr3.

. CHOP binds to the promoter of Tbet and acts a suppressor in functional reporter assays.

. Deletion or suppression of Ddit3 expression improves B16 melanoma anti-tumor response.

Comments:

Figure 1. Chop transcript was observed to be elevated in bulk CD8CD45 positive T cells from s.c. tumor models and from spleens. What is frequency of CHOP⁺ cells in the TILs?

Figure 2: CD4 Cre mice were used in the Eif2ak3flox/flox mice, yet the authors studied the CD8 T cell lineage. Shouldnt excision be restricted only to the CD4 lineage? Please clarify.

Figure 2H: What is the question behind costimulation of T cells in the presence of ascites? What factor in ascites is responsible for the upregulation of pPERK or provoking ROS production. This is not clear in the manuscript.

The sequence of data in this figure suggests that ROS induction in T cells is key for the upregulation of pPERK. What is the role of ascites in inducing T cell ROS? Why is the CD3/CD28 induction of pPERK so much less when compared to Fig 2D?

Fig3A: what fold increase and FDR was used to categorize genes increasing or decreasing in the Ddit3^{-/-} T cells?

Figure 5: The anti-tumor effect of Ddit3 deletion was measurable but modest in the B16 model. Was the same observed in the other syngeneic tumor models?

The authors suggest that PERK is a critical upstream regulator of CHOP. Was ATF4 upregulated in their T cells? Is ATF4 the critical TF that regulates CHOP in this system?

Fig 6: Its interesting that the authors observed a significant anti-tumor effect using the adoptive cellular therapy model against the melanoma tumor antigen gp100, in which pre-activated anti-gp10025-33 transgenic CD90.1⁺ Pmel T-cells were transferred into CD90.2⁺ congenic mice bearing established B16 melanoma tumors and similarly using anti-sense oligos against Ddit3. Yet the tumors were not completely eradicated and ultimately escape. Do the authors know anything about the nature of the resistant tumor or other modifications of the inflammatory infiltrate in the resistant recurrent tumor mass?

Have the authors used PERK inhibitors in their in vivo studies? Do they recapitulate their findings seen genetically with deletion of Ddit3?

Does loss of CHOP lead to resilient T cells that are impervious to cell death signals?

Lastly, what is the role of other stress related kinases in mediated pEIF2 α and CHOP induction: eg. HRI, PKR, and GCN2?

Minor comments:

The author should make clear that clone 9C8 and R-20 are anti-CHOP hybridoma antibodies. While described in the methods, it is confusing in the text what "clone" refers to. Simple clarification would help with hunting around in the text to figure this out.

Line 134 "and 12 controls" and in the legend of the figure it's 10 controls.

Line 237 "To further understand the role of a decreased T-bet in the effects induced by Chop deletion" but if Chop represses Tbx21 promoter activity when it is deleted T-bet increases.

Assessment: Excellent paper, well written with important potential impact. Please consider this manuscript for publication pending response to the queries raised in this review.

Reviewer #1: Immunometabolism

1. Figure 1E,F. The authors maintained that CHOP nuclear localization was enhanced in CD8⁺ TILs from patients with ovarian carcinoma. However, the images provided in the Figure do not show nuclear localization; rather CHOP was predominantly found in the cytoplasm. The authors need to show more convincing images (higher magnification?) that demonstrate CHOP nuclear localization.

We agree with the reviewer on the importance of clarifying the subcellular distribution of CHOP in CD8⁺ TILs. Thus, we used a higher magnification of the immunofluorescence of CHOP in CD8⁺ T-cells infiltrating ovarian tumors to illustrate its localization. Specific staining with 2 independent antibodies showed the elevated distribution of CHOP in the cytosol and the nucleus of CD8⁺ TILs, compared to CD8⁺ T-cells from healthy ovaries (Figure 1E, and Figure S2C-D). Next, we correlated the non-subjective values of nuclear and cytosolic CHOP in CD8⁺ TILs from ovarian tumors with clinical responses. The increased expression of nuclear CHOP in CD8⁺ TILs correlated with lower overall survival and unsuccessful ovarian tumor debulking, whereas cytosolic CHOP was only associated with unsuccessful cytoreductive surgery (Figure 1F-H and Figure S2E, F-G). We argue that the enhanced regulatory effect of the nuclear T cell-Chop in ovarian cancer is potentially mediated by its negative effect in the activity of primary

transcriptional mediators of T-cell function, including NF κ B and NFAT (Hayashi et al., 2013), as well as the repression of *Tbx21* (T-bet) transcription (Figure 4). We highlighted these important targets of T-cell suppression by Chop in the new version of the discussion (Lines 384-392). Furthermore, please find the related answer for Reviewer 2-Comment 13, in which we address the overall increased expression of CHOP in the whole ovarian tumor vs. CD8⁺ TILs.

2. Figure 2A,C,D,H,I. These immunoblot images lack proper quantification. The authors need to add graph summaries of these subfigures.

In the new version of the manuscript, we included the suggested quantitation of the immunoblot images and added the normalized densitometry results merging all the experimental repeats (Figure 2A, C-E, H-I, and Figure S4A, S5B, D).

3. Figure 2B. The authors examined Chop expression in proliferating cells, however in the figure legend it was stated that the authors probed for expression in CFSE⁺ cells. The authors should only assess Chop MFI in CFSE low cells.

We thank the reviewer for the suggestion. The recommended comparison of Chop in CFSE^{low}-proliferating T-cells is challenging because the transferred T-cells in non-vaccinated mice only show a minimal CFSE dilution as the result of their homeostatic division. However, to address the point of the reviewer, we compared the expression of Chop in the adoptively transferred CFSE^{diluting}-proliferating Pmel T-cells from gp100-vaccinated mice (activation-driven T-cell proliferation) vs. non-vaccinated controls (homeostatic T-cell division). Results show higher levels of Chop in the proliferating Pmel T-cells from vaccinated mice, compared to those from non-vaccinated cohorts (Figure S3D), suggesting the elevated expression of Chop under activation-driven proliferation. Additionally, we have re-formatted the Figure 2B to emphasize the comparison of the Chop levels and the frequency of Chop⁺ cells in the adoptively transferred CFSE-labeled Pmel CD8⁺ T-cells from mice with or without gp100 vaccination (Lines 168-174).

4. Figure 2C. The authors need to demonstrate whether PERK phosphorylation was also enhanced following TUDCA or thapsigargin treatment.

The new Figure 2C shows the effects of TUDCA and thapsigargin in the phosphorylation of Perk in activated CD8⁺ T-cells. In agreement with the postulated upstream role of Perk in the induction of Chop, we found an increased phosphorylation of Perk and elevated levels of Chop after treatment of the stimulated T-cells with thapsigargin. Conversely, the exposure of primed T cells to the ER stress scavenger, TUDCA, decreased the expression of phospho-Perk and Chop.

5. Figure 2D. CHOP induction following PERK activation is mediated by the transcription factor ATF4. Is ATF4 protein abundance similarly enhanced in mouse and human T cells following stimulation? The authors need to probe for ATF4 expression using a similar time-course.

We thank the Reviewer for this suggestion. Our new results show similar expression kinetics for Atf4, Chop, and phospho-Perk in murine and human activated T-cells (Figure 2A, D). In

addition, we noticed a significant decrease in *Ddit3* mRNA in stimulated T-cells transfected with a pool of specific siRNA targeting *Atf4*, as compared to controls (Figure S4D-E), demonstrating the primary role of the induction of *Atf4* in the upregulation of Chop in primed T-cells.

6. Figure 2H. The authors show that addition of ascites from tumor bearing mice increases PERK pathway activity and Chop expression. However, in this figure, Chop induction following activation for 48 h was noticeably lower compared to Fig 2A. The authors need to show a graphic quantification of these experiments to fully appreciate the effect of ascites on PERK pathway activity. See comment relating to Figure 2A,C,D,H,I.

We agree with the reviewer on the need of this clarification. The pointed changes in the intensity of Chop in the immunoblots from Figures 2H and 2A were the result of distinct exposure times, as well as changes in the amount of the loaded protein. As requested by the reviewer, we have re-run and scanned the figures from similar exposure times and quantitated the blot intensity from Figures 2A, C, D, H, I, S4A, and S5B, D. These results have replaced the previous illustrations.

7. Sup Figure 4C. GSEA of WT and Ddit3^{-/-} T cells revealed that CHOP deletion promoted the expression of OXPHOS and glycolysis genes. Since these two pathways are important for T cell expansion and effector function, the authors need to examine the metabolic signature of primed Ddit3^{-/-} CD8⁺ T cells compared to wild type controls by directly probing for glycolysis and mitochondrial respiration.

This is an excellent suggestion that has opened new avenues for our research. To monitor the metabolic signature of *Ddit3*-null vs. control CD8⁺ T-cells, we measured the mitochondrial oxidative capacity and the glycolytic potential by extracellular flux analyzer Seahorse. Results show an elevated mitochondrial oxidative phosphorylation and glycolytic activity potential in activated *Ddit3*-null CD8⁺ T-cells, compared to wild type controls, suggesting that the elimination of Chop promotes metabolic fitness in CD8⁺ T-cells. These findings have been illustrated in the new Figure 3F-G and a description of the results added in Lines 229-233.

8. Figure 4A. The authors examined the role of Chop in the regulation of T-bet expression. However, in this subfigure T-bet protein abundance at 48 h post-activation is considerably lower in wt versus Ddit3^{-/-} CD8⁺ T cells, which is in stark contrast to previously published findings from Anjana Rao's group and other laboratories (Cruz-Guilloty, JEM, 2009). The authors need to reconcile these data with past findings by performing a time course examination of T-bet protein abundance (similar to Figure 2A for Chop) to highlight the inverse relationship between T-bet and Chop.

We apologize for the unclear presentation of the results showing a low basal expression of T-bet in primed wild type CD8⁺ T-cells, which was the result of a short time exposure. To better illustrate the effect of Chop in the expression of T-bet in activated CD8⁺ T-cells, we have completed the suggested time course examinations. An augmented expression of T-bet was found in stimulated *Ddit3*-null CD8⁺ T-cells, compared to controls (Figure 4A), indicating the inverse relationship between Chop and T-bet in stimulated T-cells.

9. Figure 4B,C. The authors demonstrated that the effector maturation program is enhanced in *Ddit3* deficient CD8⁺ T cells. However, the authors have not appropriately examined the differentiation status of these cells. Are *Ddit3*^{-/-} CD8⁺ T cells differentiating into terminally differentiated T cells? It was shown in Fig 3I that these cells display an effector memory phenotype, as evidenced by an enhanced potential to differentiate into CD44⁺CD62L^{neg} cells. However, these are not the appropriate markers to evaluate the differentiation status of newly primed cells. The authors should establish whether differentiation of *Chop* deficient T cells into effector/memory subpopulations is altered by using *KLRG1* and *CD127* as surface markers. These two markers should appropriately mark early differentiation into short-lived effector and memory T cells.

To address this important point, we tested the expression of *KLRG1* and *CD127* in: 1) *Ddit3*-null and control CD8⁺ T-cells activated with anti-CD3/CD28; and 2) CD8⁺ TILs from wild type and T-cell conditional-*Ddit3*-ko mice bearing B16 tumors. As previously reported (Robbins et al., 2003), we did not find a significant expression of *KLRG1* in the cultured activated CD8⁺ T-cells. Conversely, we observed an elevated frequency of *KLRG1*^{high} *CD127*^{low} cells in CD8⁺ TILs from T cell-conditional *Ddit3*-null mice, compared to control counterparts (Figure 5E). Thus, deletion of *Chop* induced the accumulation of late effector CD8⁺ T-cells. This is relevant as previous results showed that the elevated expression of T-bet promoted the accumulation of *KLRG1*^{high} *CD127*^{low} CD8⁺ T-cells (Badovinac and Harty, 2007; Joshi et al., 2007). Our future studies will test the role of the elevated T-bet in *Ddit3*-null mice in the expansion of this population.

10. Figure 4D-H. The authors elegantly examined the importance of *Chop* in the repression of T-bet expression. However, in Figure 3B, expression of 3 other transcription factors –*Id2*, *Prdm1* and *Eomes*– that regulate the differentiation of activated CD8⁺ T cells were also enhanced in *Ddit3* deficient cells. The authors need to determine whether protein abundance of all three transcription factors is similarly enhanced as T-bet in *Ddit3*^{-/-} cells, and whether these proteins are directly regulated by *Chop*, by analyzing for putative *Chop* binding sites in their respective promoters using the ChampionChip database.

We agree with the reviewer on the importance of identifying the effect of *Chop* on the expression of additional key mediators of CD8⁺ T-cell effector responses and differentiation. Although we observed higher levels of *Eomes* in the RNAseq analysis (Figure 3B), we did not detect a significant upregulation of *Eomes* protein in stimulated CD8⁺ T-cells from wild type or *Ddit3* deficient mice (Figure S7A). This limited expression of *Eomes* protein in *in vitro* activated CD8⁺ T-cells cultured in the absence of IL-2 has been previously reported (Cho et al., 2013). In addition, new time course protein studies showed a similar expression of *Id2* and *Prdm1*/*Blimp1* in stimulated *Ddit3* deficient and wild type CD8⁺ T-cells (Figure S7A). Furthermore, prediction analysis developed using the ChampionChIP, TFTG-BD and ChIPBase v2.0 databases indicated the lack of consensus binding sites for *Chop* on the promoter regions of *Id2*, *Prdm1*, and *Eomes* genes. Together, these results show the direct regulation of T-bet by *Chop* as a potential specific signal regulating the effector activity and differentiation of CD8⁺ T-cells.

11. Figure 5B,C. Polyfunctional T cells have been demonstrated to be important for control of many cancer types (Zhao, E., Nat Imm, 2016). Are the frequencies of polyfunctional (IFN-gamma and TNF double positive) cells also increased in Chop deficient cells?

To address this important comment, we tested the dual production of IFN γ and TNF α in CD8⁺ TILs from flox and T cell-conditional *Ddit3*-deficient mice bearing B16 tumors. Results show a dual augmented production of IFN γ and TNF α in CD8⁺ TILs from T cell-conditional *Ddit3*-deficient mice, as compared to CD8⁺ TILs from flox controls (Figure 5B-C). Thus, deletion of Chop in T-cells promotes the expansion of polyfunctional tumor-associated CD8⁺ T-cells.

12. Figure 5D-G. Mislabeled in text and figure.

We apologize for this mistake. The legends for the Figure 5D-G have been reviewed and the appropriate corrections have been included in the new version of the manuscript.

Reviewer #2 : Cancer immunology

13. CHOP imaging on ovarian tumor samples from patients shows that CHOP is expressed everywhere in the tumor and not specifically in the CD8 T cells. The authors state that nuclear CHOP level in CD8 correlates with a better survival. Is it not just the general level of CHOP in the entire tumor and not specifically in the CD8 T cells that is correlated with survival? Furthermore the images shown in Figure 1 do not convince me that CHOP is intra-nuclear in the CD8 T cells. I would like to see pictures of what the authors consider as high nuclear content versus a low.

We thank the reviewer for the opportunity to clarify this relevant point. We observed a broad upregulation of CHOP in advanced ovarian tumors compared to healthy ovarian tissues (Figure S2A-B, Figure for Reviewers 1A-B), indicating that the expression of CHOP was not limited to CD8⁺ TILs. In order to test the significance of the CHOP expression in ovarian tumors and ovarian CD8⁺ TILs (cytosolic or nuclear), non-subjective studies were completed in single-fluorochrome scanned tissue microarray histology slides (APERIO ScanScope-FL) using the Definiens Tissue Studio v4.2 analysis software (Methods for imaging). For the analysis in CD8⁺ TILs, nuclear DAPI and CD8-FITC channels were used to segment CHOP-Cy5 in the nucleus and cytoplasm, respectively, based on algorithms of intensity and size constraints. Using studies with 2 independent antibodies, we found that the expression of Chop in the whole ovarian tumor tissue did not impact the overall survival or the success of the cytoreductive surgery (Figure for Reviewers 1C-F). Conversely, the augmented expression of nuclear CHOP in CD8⁺ TILs correlated with poor overall survival and unsuccessful ovarian tumor debulking in patients with advanced ovarian carcinoma (Figure 1F-H). Thus, the nuclear expression of CHOP in CD8⁺ TILs correlated with worse clinical outcome in patients with advanced ovarian tumors. Furthermore, to clarify the subcellular distribution of CHOP in CD8⁺ TILs, we provide new magnification slides showing the expression of Chop in the cytosol and nucleus of CD8⁺ TILs, as

well as representative results illustrating a spectrum of Chop^{high} vs. Chop^{low} CD8⁺ TILs in the ovarian carcinoma TMA slides (Figure 1E, Figure S2A, and Figure for Reviewers 1G).

14. Concerning figure 2, most of the readouts have been carried out with western blots. There are no quantification of the data. Furthermore, the anti-CHOP Ab seems to work properly for flow cytometry according to figure 1. The results obtained by western blots have to be confirmed by flow cytometry for CHOP expression.

As suggested by the reviewer, we semi-quantitated the immunoblotting band intensities and plot the results in new figures that summarize the independent repeats (Figure 2 and Figure S4-5). Additionally, we validated the Chop immunoblotting time course studies by flow cytometry. Similar to the western blot results, we found a time-related increase in Chop expression in activated CD8⁺ T-cells by flow cytometry (Figure S3A).

15. In general, for all the figures, I would prefer to see dots and not plain columns for the histograms. This way we can see how many mice or samples have been used. Some statistical analyses have been performed with only 3 replicates per condition. The number of samples should be increased for those. It occurred in several histograms.

We agree with the reviewer on the need of this change. Thus, we have reformatted the plain column figures throughout the manuscript for scatter plot with bar graphs (Figure 1-6 and Figure S1-9). This change addresses the comment of the reviewer and allows a better illustration of our

findings. In addition, we have increased the number of experimental repeats for several of the histogram-based studies and updated the legends description and the statistical analysis.

16. For figure 3H and D, only one dot plot is shown for each. Combined results for several experiments have to be shown too.

We apologize for the lack of clarity as we aimed to illustrate a representative result. We now include figures that show the merged repeats for the experiment and that indicate the higher cytotoxic potential (Figure 3J, right) and the differentiation of the activated *Ddit3*-null CD8⁺ T-cells and controls into CD62L^{neg} CD44⁺ cells (Figure S6F).

17. For figure 4H there is no mention about the number of experiments done.

A total of 3 repeats were completed to test the functional effect of a low T-bet in the inhibition of IFN γ by Chop (Figure 4H). An additional graph summarizing the repeats is also included.

18. In the legend for figure 5B, it is mentioned that compiled results for 5 mice are shown... I don't have these results on the figure.

We apologize for this mistake. The final total number of independent testing for the Figures 5B-C is 10 tumors. Therefore, we have corrected the number of mice in the legends.

19. There are some mislabeled figures in the text compared to the figures (Figure 5D instead of 5F and 5E instead of G).

We thank the reviewer for pointing this error. We have corrected all these highlighted mistakes and proofread the manuscript.

20. For figure 5F, it is indicated that only 3 mice per group have been used. This number has to be increased. Similarly, experiment for figure 5G has to be repeated (4 mice per group).

As recommended, we have repeated the experiments from Figure 5 and increased the number of mice. The total number of mice for the Figure 5F is 6/group and for Figure 5G is 8/group.

21. In figure 6C the number of mice shown is not indicated. Similar remark for figure 6E.

We apologize for the lack of clarity. A more clear description for the Figures 6C and E have been added in the new version of the legends and include the number of repeats (Figure 6C=7; and Figure 6E= 4).

22. For figure 6F there are 4 mice per group only. This experiment has to be repeated.

We repeated the experiment to increase the number of mice (8 per group), showing a similar anti-tumor effect. A merged illustration of the experiments is included in the new Figure 6F.

Reviewer #3: Ovarian Cancer

23. *Figure 1. Chop transcript was observed to be elevated in bulk CD8-CD45 positive T cells from s.c. tumor models and from spleens. What is frequency of CHOP+ cells in the TILs?*

We thank the reviewer for the important inquiry. Thus, we determined by flow cytometry the frequency of Chop⁺ cells among the CD8⁺ T-cells from tumors and spleen of mice bearing B16, 3LL, or ID8-Vegf-a tumors, as well as in CD8⁺ T-cells from tumors and peripheral blood from patients with advanced ovarian carcinoma. In all tested tumor models, we observed an increased frequency of Chop⁺ cells within the CD8⁺ TILs, compared to spleen or peripheral blood CD8⁺ T-cells (**Figure S1B and D**, B16 (62.7% vs. 0.9%), 3LL (19.7% vs. 2.1%), ID8-Vegf-a (44% ± vs. 3.6%) and ovarian carcinoma patients (69.1% vs. 8.0%)). Additionally, an increased accumulation of CHOP⁺ CD8⁺ TILs was detected in human ovarian tumors using non-subjective imaging approaches (**Figure 1F and S2C-D**). Thus, results show the increased frequency of CHOP⁺ CD8⁺ T-cells in various tumor models.

24. *Figure 2: CD4 Cre mice were used in the Eif2ak3flox/flox mice, yet the authors studied the CD8 T cell lineage. Shouldn't excision be restricted only to the CD4 lineage? Please clarify.*

We appreciate the opportunity to clarify this aspect. The expression of CD4-Cre recombinase induces the floxed-gene recombination in CD8⁺ T-cells by targeting the CD4⁺ CD8⁺ double-positive precursor T-cells in the thymus (Sawada et al., 1994). Thus, the CD4-Cre system will induce the gene excision in both CD4⁺ and CD8⁺ T-cells. Our results showed that depletion of CD8⁺ T-cells prevented the anti-tumor effects observed in T cell-conditional *Ddit3*-deficient mice (**Figure 5F**), suggesting the key role of CD8⁺ T-cells. Although these results do not rule out a potential contribution of Chop on CD4⁺ T-cells, we emphasized in this manuscript on the importance of Chop in CD8⁺ T-cell anti-tumor immunity.

25. *Figure 2H: What is the question behind stimulation of T cells in the presence of ascites? What factor in ascites is responsible for the upregulation of pPERK or provoking ROS production. This is not clear in the manuscript.*

The rationale for the stimulation of T lymphocytes in the presence of the ascites is to mimic the immunosuppressive effects induced by the ovarian tumor microenvironment. In regards to the signals driving the production of ROS in ascites-exposed CD8⁺ T-cells, our new results suggest the potential role of CD38 as a mediator. Thus, exposure of primed CD8⁺ T-cells with ovarian ascites resulted in a heightened expression of CD38 (**Figure for Reviewers 2A**). Notably, knockdown of *Cd38* in CD8⁺ T-cells blunted the induction of ROS by the ovarian ascites (**Figure for Reviewers 2B**), suggesting a major effect of CD38 in the activation of ROS-driven ER stress. Despite these promising results, we have been unable to identify yet a specific mediator driving the expression of CD38 and Chop in CD8⁺ TILs. Our recently published collaboration showed the relevance of the alterations in glucose uptake and protein *N*-linked glycosylation as drivers of ER stress and Xbp1 hyper-activation in CD4⁺ TILs (Song et al., 2018). Based on these results, our ongoing collaborative studies with Drs. Mehrotra and Cubillos-Ruiz aim to elucidate the mechanistic crosstalk between the glucose uptake, the induction of CD38, and the activation of

ROS-driven ER stress in CD8⁺ TILs. Because the preliminary status of the results and the current scope of our manuscript in the effects of Chop, we would like to respectfully ask the reviewer and the editor for the opportunity to develop these studies as an independent story.

26. The sequence of data in this figure suggests that ROS induction in T cells is key for the upregulation of pPERK. What is the role of ascites in inducing T cell ROS? Why is the CD3/CD28 induction of pPERK so much less when compared to Fig 2D?

Please find our response on the role of the ascites in the induction of T cell-ROS in our previous answer (Reviewer 3, question 25). Also, the differences in the phospho-Perk intensities in the Figures 2D, H, and I were the result of distinct exposure times and changes in the loaded amount of protein (Reviewer 1, question 6). Thus, to better illustrate our data, we re-run and scanned the figures from similar exposure times and quantitated the blot intensities (Figure 2A, C-E, H-I, and Figure S4A, S5B, D).

27. Fig3A: what fold increase and FDR was used to categorize genes increasing or decreasing in the *Ddit3*^{-/-} T-cells?

Figure 3A was generated by gene set enrichment analysis (GSEA) to show the signaling pathway signatures enriched in *Ddit3*^{-/-} vs. control Pmel T-cells. Transcripts obtained from differential expression (DE) analysis were ranked based on a metrics score derived from both fold change (ranking from log₂ of -4.02 to 0.82) and FDR values of each DE gene so that the most up-regulated genes are at the top of the list, the most down-regulated genes at the bottom, and the genes with no change in the middle. Next, the ranked gene list was used to run GSEA to elucidate enriched pathway signatures. Thus, the enriched effector T-cell signature shown in Figure 3A had a FDR value of 0.00369. A description of this process is now included in the Methods (Lines 554-576).

28. *Figure 5: The anti-tumor effect of Ddit3 deletion was measurable but modest in the B16 model. Was the same observed in the other syngeneic tumor models?*

We thank the reviewer for this suggestion. In agreement with the anti-tumor effects found in the B16 tumor model, we found a significant delay in tumor growth and an extended survival in T cell-conditional *Ddit3*-null mice bearing s.c. MCA-38 cells or i.p. ID8-*Defb29/Vegf* tumors, respectively, as compared to floxed controls (Figure S8B-C).

29. *The authors suggest that PERK is a critical upstream regulator of CHOP. Was ATF4 upregulated in their T cells? Is ATF4 the critical TF that regulates CHOP in this system?*

Please see the answer for Reviewer 1-question 5. Briefly, our results show similar expression kinetics of Atf4, Chop, and phospho-Perk in primed T-cells (Figure 2A, D). Moreover, an impaired upregulation of *Ddit3* mRNA was found in activated CD8⁺ T-cells transfected with specific *Atf4*-siRNA pools, compared to mock-siRNA carrying controls (Figure S4D-E). Thus, as anticipated by the reviewer, the upregulation of Atf4 by Perk plays a major role in induction of Chop in stimulated T-cells.

30. *Fig 6: It's interesting that the authors observed a significant anti-tumor effect using the adoptive cellular therapy model against the melanoma tumor antigen gp100, in which pre-activated anti-gp10025-33 transgenic CD90.1+ Pmel T-cells were transferred into CD90.2+ congenic mice bearing established B16 melanoma tumors and similarly using anti-sense oligos against Ddit3. Yet the tumors were not completely eradicated and ultimately escape. Do the authors know anything about the nature of the resistant tumor or other modifications of the inflammatory infiltrate in the resistant recurrent tumor mass?*

To address this important question, we monitored the frequency of highly immunosuppressive populations in tumors from B16-bearing mice that received Pmel T-cells transfected with anti-sense oligonucleotides (ASO) targeting *Ddit3* or mock-ASO. An augmented frequency of tumor-associated macrophages (TAM), a cellular population known to be highly immunosuppressive (Rodriguez et al., 2004), was detected in B16-bearing mice undergoing adoptive-T cell therapy (Figure for Reviewers 3). Notably, the accumulation of TAM was further augmented in tumors from mice receiving *Ddit3*-ASO Pmel T-cells, compared to those transferred with mock-ASO T-cells (Figure for Reviewers 3), indicating that the development of TAM in mice receiving *Ddit3*-ASO Pmel T-cells could eventually counter the augmented anti-tumor T-cell activity.

Figure for Reviewers 3. Tumor-associated macrophages (CD45⁺ CD11b⁺ F4/80⁺) in tumors from mice bearing B16 melanoma and that were treated or not with CD8 Pmel T cells pre-incubated with gp100 and control ASO (blue) or Chop ASO (red). Tumors were collected 5 days after transfer.

31. Have the authors used PERK inhibitors in their in vivo studies? Do they recapitulate their findings seen genetically with deletion of *Ddit3*?

Results presented below show the anti-tumor effect of the Perk inhibitors, AMG-44 (highly specific Perk inhibitor) or GSK-2606414 (Perk and RIPK1 kinase inhibitor) (Rojas-Rivera et al., 2017; Smith et al., 2015) in B16 tumor-bearing mice. A delayed tumor growth, correlating with an increased tumor infiltration of IFN γ -expressing CD8⁺ T-cells, was observed in mice treated with each of the Perk inhibitors, compared to vehicle-treated controls (Figure for Reviewers 4A-D). Thus, the overall inhibition of Perk mimics the immunogenic effects found in T cell-conditional *Ddit3*-null mice. One of our current non-overlapping studies aims to discriminate the tumor immunity effects of Perk in cancer cells, CD8⁺ T-cells, and myeloid subsets. Therefore, if the reviewers and editor allow us, we would like to use these results for this focused project.

32. Does loss of CHOP lead to resilient T cells that are impervious to cell death signals?

Although the upregulation of Chop in proliferating T-cells did not correlate with a significant induction of apoptosis, a slight decrease in spontaneous and staurosporine-enforced expression of the apoptosis marker Annexin V was detected in stimulated *Ddit3* deficient CD8⁺ T-cells, compared to controls (Figure S6D-E). Thus, results suggest a potential survival benefit in CD8⁺ T-cells upon elimination of Chop. We have included comments pointing on these findings in the results section (Lines 227-229).

33. Lastly, what is the role of OTHER stress related kinases in mediated pEIF2 α and CHOP induction: eg. HRI, PKR, and GCN2?

We consider this to be a fundamental point and thank the reviewer for the comment. Thus, we tested the expression of *Ddit3* mRNA in activated T-cells transfected with a pool of specific

siRNA targeting HRI (*Eif2ak1*), PKR (*Eif2ak2*), Perk (*Eif2ak3*), or GCN2 (*Eif2ak3*) (Accell SMART-pool siRNA, Dharmacon) (Figure S4G). Although the knockdown of *Eif2ak3* had the highest effect in the prevention of *Ddit3* upregulation, we noted that silencing of *Eif2ak2* and *Eif2ak4* also inhibited the induction of *Ddit3* in activated T-cells (Figure S4H). Thus, although the activation of the integrated stress responses (ISR) by Perk is the most relevant Chop-driving signal in stimulated T-cells, the potential activation of the ER stress-independent kinases Gcn2 and Pkr could also regulate the expression of Chop in CD8⁺ T-cells.

Minor comments:

34. The authors should make clear that clone 9C8 and R-20 are anti-CHOP hybridoma antibodies. While described in the methods, it is confusing in the text what “clone” refers to. Simple clarification would help with hunting around in the text to figure this out.

We appreciate the opportunity to clarify the source of the Chop antibodies. We have modified the description of the antibodies in the narrative and the figures to clarify that the clone 9C8 is a hybridoma-produced monoclonal IgG2b antibody, whereas the R-20 is a primary rabbit polyclonal antibody. We appreciate the comment and hope that our answer satisfies the request.

35. Line 134 “and 12 controls” and in the legend of the figure it’s 10 controls.

The typo has been corrected in the legends. The correct number of controls is 12.

36. Line 237 “To further understand the role of a decreased T-bet in the effects induced by Chop deletion” but if Chop represses Tbx21 promoter activity when it is deleted T-bet increases.

We thank the reviewer for pointing this mistake. We have replaced the sentence for “To further understand the role of a decreased T-bet in the effects induced by Chop upregulation” (lines 275-276)

Thanks again for your consideration of this manuscript; please let me know if there is additional information needed to proceed with this process.

Sincerely,

Paulo C. Rodriguez, PhD
Associate Member,
Moffitt Cancer Center and Research Institute,
Department of Immunology,
12902 Magnolia Drive-MRC-Annex 2067E,
Tampa, FL 33612; Phone: 813-745-1457

References

- Badovinac, V.P., and Harty, J.T. (2007). Manipulating the rate of memory CD8+ T cell generation after acute infection. *J Immunol* *179*, 53-63.
- Cho, J.H., Kim, H.O., Kim, K.S., Yang, D.H., Surh, C.D., and Sprent, J. (2013). Unique features of naive CD8+ T cell activation by IL-2. *J Immunol* *191*, 5559-5573.
- Hayashi, K., Jutabha, P., Endou, H., Sagara, H., and Anzai, N. (2013). LAT1 is a critical transporter of essential amino acids for immune reactions in activated human T cells. *J Immunol* *191*, 4080-4085.
- Joshi, N.S., Cui, W., Chandele, A., Lee, H.K., Urso, D.R., Hagman, J., Gapin, L., and Kaech, S.M. (2007). Inflammation directs memory precursor and short-lived effector CD8(+) T cell fates via the graded expression of T-bet transcription factor. *Immunity* *27*, 281-295.
- Robbins, S.H., Terrizzi, S.C., Sydora, B.C., Mikayama, T., and Brossay, L. (2003). Differential regulation of killer cell lectin-like receptor G1 expression on T cells. *J Immunol* *170*, 5876-5885.
- Rodriguez, P.C., Quiceno, D.G., Zabaleta, J., Ortiz, B., Zea, A.H., Piazuelo, M.B., Delgado, A., Correa, P., Brayer, J., Sotomayor, E.M., *et al.* (2004). Arginase I production in the tumor microenvironment by mature myeloid cells inhibits T-cell receptor expression and antigen-specific T-cell responses. *Cancer Res.* *64*, 5839-5849.
- Rojas-Rivera, D., Delvaeye, T., Roelandt, R., Nerinckx, W., Augustyns, K., Vandenabeele, P., and Bertrand, M.J.M. (2017). When PERK inhibitors turn out to be new potent RIPK1 inhibitors: critical issues on the specificity and use of GSK2606414 and GSK2656157. *Cell Death Differ* *24*, 1100-1110.
- Sawada, S., Scarborough, J.D., Killeen, N., and Littman, D.R. (1994). A lineage-specific transcriptional silencer regulates CD4 gene expression during T lymphocyte development. *Cell* *77*, 917-929.
- Smith, A.L., Andrews, K.L., Beckmann, H., Bellon, S.F., Beltran, P.J., Booker, S., Chen, H., Chung, Y.A., D'Angelo, N.D., Dao, J., *et al.* (2015). Discovery of 1H-pyrazol-3(2H)-ones as potent and selective inhibitors of protein kinase R-like endoplasmic reticulum kinase (PERK). *J Med Chem* *58*, 1426-1441.
- Song, M., Sandoval, T.A., Chae, C.S., Chopra, S., Tan, C., Rutkowski, M.R., Raundhal, M., Chaurio, R.A., Payne, K.K., Konrad, C., *et al.* (2018). IRE1alpha-XBP1 controls T cell function in ovarian cancer by regulating mitochondrial activity. *Nature* *562*, 423-428.

Reviewers' Comments:

Reviewer #1:

Remarks to the Author:

All my major concerns with the original version of the manuscript were addressed by the authors. Interrogating the impact of the unfolded protein response on immune cell function is timely and interesting.

Reviewer #2:

Remarks to the Author:

The comment of the various reviewers have been thoroughly addressed. I now recommend the publication of this revised article.

Reviewer #3:

Remarks to the Author:

This manuscript is now ready for publication and has addressed the substantial number of issues raised by each of the reviewers.

One minor point:

1. "self-phosphorylation" of the protein kinase R (PKR)-like ER kinase (page 6), should read autophosphorylation.